# Tumor histoculture captures the dynamic interactions between tumor and immune components in response to anti-PD1 in head and neck cancer

Nandini Pal Basak[1,7], Kowshik Jaganathan[1,7], Biswajit Das [1],
Oliyarasi Muthusamy[1], Rajashekar M[1], Ritu Malhotra[1], Amit Samal[1], Moumita Nath[1],
Ganesh MS[2], Amritha Prabha Shankar[2], Prakash BV[3], Vijay Pillai[4], Manjula BV[5],
Jayaprakash C[6], Vasanth K[1], Gowri Shankar K[1], Sindhu Govindan[1], Syamkumar V[1],
Juby[1], Koushika R[1], Chandan Bhowal[1], Upendra Kumar[1], Govindaraj K[1],
Mohit Malhotra [1] & Satish Sankaran [1] ✉

Dynamic interactions within the tumor micro-environment drive patient response to immune checkpoint inhibitors. Existing preclinical models lack true representation of this complexity. Using a Head and Neck cancer patient derived TruTumor histoculture platform, the response spectrum of 70 patients to anti-PD1 treatment is investigated in this study. With a subset of 55 patient samples, multiple assays to characterize T-cell reinvigoration and tumor cytotoxicity are performed. Based on levels of these two response parameters, patients are stratified into five sub-cohorts, with the best responder and non-responder sub-cohorts falling at extreme ends of the spectrum. The responder sub-cohort exhibits high T-cell reinvigoration, high tumor cytotoxicity with T-cells homing into the tumor upon treatment whereas immune suppression and tumor progression pathways are pre-dominant in the non-responders. Some moderate responders benefit from combination of anti-CTLA4 with anti-PD1, which is evident from better cyto-toxic T-cell: T-regulatory cell ratio and enhancement of tumor cytotoxicity. Baseline and on-treatment gene expression signatures from this study stratify responders and non-responders in unrelated clinical datasets.

Immune cell mediated tumor killing is brought about by a series of regulated biological events involving the cytotoxic T Lymphocytes (CTLs)[1]. T-cell reinvigoration, followed by cytotoxic T-cell infiltration into the tumor nest, release of Interferon gamma (IFNg) and cytolytic proteins (GranzymeB and Perforin) finally leads to tumor cell killing[1].

CTL activation is followed by its exhaustion, mediated by other regulatory immune cell sub-populations via multiple immune check point inhibitor pathways to maintain immune system homeostasis and pre-vent autoimmunity. These inhibitory pathways are hijacked by the tumor cells rendering the intra-tumoral CTLs dysfunctional and

[1]Farcast Biosciences India Pvt. Ltd, Bangalore, Karnataka, India. [2]Vydehi Institute of Medical Sciences & Research Centre, Bangalore, Karnataka, India. [3]Sri Lakshmi Multi-Speciality Hospital, Bangalore, Karnataka, India. [4]Mazumdar Shaw Medical Center, Narayana Health, Bangalore, Karnataka, India. [5]Bangalore Baptist Hospital, Bangalore, Karnataka, India. [6]DBR & SK Super Speciality Hospital, Tirupati, Andhra Pradesh, India. [7]These authors contributed equally: Nandini Pal Basak, Kowshik Jaganathan. ✉e-mail: satish.sankaran@farcastbio.com

exhausted[2]. Immune check point inhibitors like anti-PD1 and anti-CTLA4 are designed to enhance T-cell mediated cytotoxicity by blocking the PD1-PDL1 and CTLA4-CD80/86 mediated T-cell exhaustion mechanism leading to reinvigoration of T-cells[3]. Response to Immune Checkpoint Inhibitors (ICI) therapies in patients is highly variable and driven by their heterogenous tumor micro-environment components which constitute tumor, immune cells[4], and stromal cells[5]. Dynamic interactions between immune cell sub-populations and the tumor cells lead to either anti- or pro-tumorigenic effects[6] post treatment. Most preclinical response evaluation models fail to capture this complexity and heterogeneity of tumor microenvironment (TME) and hence fail to replicate in vivo response of tumor.

Tumor histocultures use tumor explants, that are cultured for short duration of time and have been shown to preserve tumor, stromal and immune cells. Such culture platforms are being explored for the prediction of clinical response to both chemotherapy and immunotherapy[7–9]. Some of the challenges with histoculture include unbiased preservation of all components of the TME; representing tumor heterogeneity in every treatment arm while maintaining equivalence across arms; choice of the right number of replicates that optimally utilizes the limited sample without compromising on arm equivalence; ability to culture explants that are large enough to capture the complex tumor immune interactions; and ability to demonstrate tumor cytotoxicity within the short culture period. Emerging evidence indicates that the ICI treatment outcome to treatment extends beyond the direct effect on T-cell reinvigoration and involves interplay between the cytotoxic T-cells (CTL), myeloid cells, and regulatory T cells (Treg) vis-a-vis their relative spatial[10] distribution. A histoculture platform which retains these immune cells in their near-native spatial contexture will thus be required to study the true effect of ICI ex vivo.

In this work, we describe the development and testing of a head and neck squamous cell carcinoma (HNSCC) patient derived TruTumor histoculture platform. The optimized explant size and number of replicates capture tumor heterogeneity and preserve various cell types in their near-native spatial contexture while maintaining their functional fidelity. Using a 70-patient sample cohort, we demonstrate a heterogenous response to Nivolumab treatment, identifying potential responders and non-responders. The platform exhibits robust T-cell reinvigoration and tumor cytotoxicity response to treatment, offering an opportunity to be developed into a clinical test for personalized patient treatment options. This study also provides insights about potential resistance markers that could be targeted to customize combination therapies for better treatment outcomes.

## Results

### Near native ex vivo human TruTrumor histoculture platform to study response to immune modulators

We developed a near-native ex vivo human HNSCC histoculture TruTumor platform using fresh, surgically resected tumors samples (Fig. 1A). Of the 98 samples recruited in this study, 28 were utilized to develop and standardize the platform. The remaining 70 samples were subjected to treatment with ICI (Fig. 1B). Based on the size of the available sample, a sub-set of these samples were analyzed using multiple assay readouts (Fig. 1B) and duplicate treatment arms. Sample demography is detailed in Table 1.

Samples with average tumor content less than 10% at baseline ($T_{BL}$) were considered non-qualified and excluded from the analysis. For HNSCC, sample attrition based on insufficient tumor content was around 19 %. A 72-hour culture period of vibratome-generated thin explants ensured minimal tissue disintegration while preserving tumor and stromal architecture (Supplementary Fig. 1A) with preservation of the intra-tumoral immune cell repertoire (Supplementary Fig. 2). There was no effect of neo-adjuvant treatment on the level of preservation of samples (Supplementary Fig. 1B). Extending culture period beyond

72 hours led to high levels of tissue discohesion and cellular pyknosis (Supplementary Fig. 1C). This period of culture, however, might not be enough to address acquired resistance to therapy. While there was a significant drop in the CD3 + T cells, proportions of cytotoxic T-cells (CD8 + ) and CD4 + T cells did not vary significantly (Supplementary Fig. 2C). Efflux of T cells from explant cultures have been reported for other explant based histocultures when cultured in absence of a matrix[9]. NK cells showed a significant drop post culture, whereas, CD68+ Macrophage population showed an increase at $T_{72}$ compared to $T_{BL}$. Notably, the ratio of M1/M2 macrophages did not show a significant change post culture. B cells, Neutrophils, and Monocytes did not vary to a great extent between $T_{BL}$ and post culture ($T_{72}$) (Supplementary Fig. 2C). We compared immune cell signatures to determine the level of immune cell infiltration or hotness of tumor samples pre- and post-culture (Supplementary Fig. 2D). Except for one sample all remaining 22 samples maintained their hotness between pre- and post-culture. To negate culture-induced effects on the explants, a control arm (without treatment with drug) was always used for normalizing response readouts. During development of this platform, we concentrated on two critical aspects, namely, optimal representation of tumor heterogeneity and maintaining arm equivalence which have not been adequately addressed in any of the histoculture assay development-related publications, to the best of our knowledge. These aspects were addressed by choosing the optimal size of the fragments, number of replicate explants per arm, and a plating strategy that ensured representation from every cross-section of the tumor sample in each arm. Seven replicates per arm was finalized after evaluating the level of variation across two parameters (total tumor content and proliferating tumor cell content) from simulated arms with comprising different replicate numbers (namely 3, 5, 7, and 10) (Supplementary Fig. 3). A seven replicate arm exhibited less than 20% CV across replicate arms and creation of the optimal number of treatment arms from the limited tumor sample quantity. Actual arm equivalence was confirmed across samples through multiple other experiments, assessing multiple parameters like tumor content (Supplementary Fig. 4A), immune content (Supplementary Fig. 4A), percentage of proliferating tumor cells (Ki67 + ) (Supplementary Fig. 4A), explant viability (Supplementary Fig. 4B), gene expression using mRNA counts (Supplementary Fig. 4C), and proportions of various immune cells (Supplementary Fig. 4D). Reproducible response to treatment with Nivolumab evaluated using IFNg release, between duplicate arms across multiple samples ($n = 8$) was observed (Supplementary Fig. 4E).

Intra-tumoral T-cells functional fidelity was established by assessing cytokine release on treatment with T-cell stimulants (anti-CD3 + IL2), and myeloid cell stimulants, (Lipopolysaccharide, LPS). We observed specific release of IFNg, GranzymeB and Perforin upon anti-CD3 + IL2 treatment and not upon LPS treatment for the same tumor sample (Fig. 1C). Increase in CD8+GranzymeB+ and CD8 + Ki67+ sub-population, CTL/Treg ratio and IFNg Gene Expression Signatures (GES) were specifically observed on anti-CD3 + IL2 treatment, whereas, IL1α + IL1β gene expression increased with LPS (Supplementary Fig. 5A and B).

### Variable T-cell reinvigoration response upon anti-PD1 treatment observed across patient samples

Between the two anti-PD1 agents, Pembrolizumab and Nivolumab, we chose Nivolumab for response evaluation. This choice was driven by high cost of Pembrolizumab and the desire to develop a clinical test for patient selection in mid and low-income country like India, for a relatively more affordable drug.

A heterogeneous IFNg release response was observed across the 70-sample cohort (Fig. 2A) on Nivolumab treatment. At a cohort level, however, a significant increase in CTL sub-population ($p < 0.05$), with even higher and significant increase in IFNg release ($p < 0.0001$) was observed on treatment with Nivolumab (Fig. 2B). Nivolumab treatment

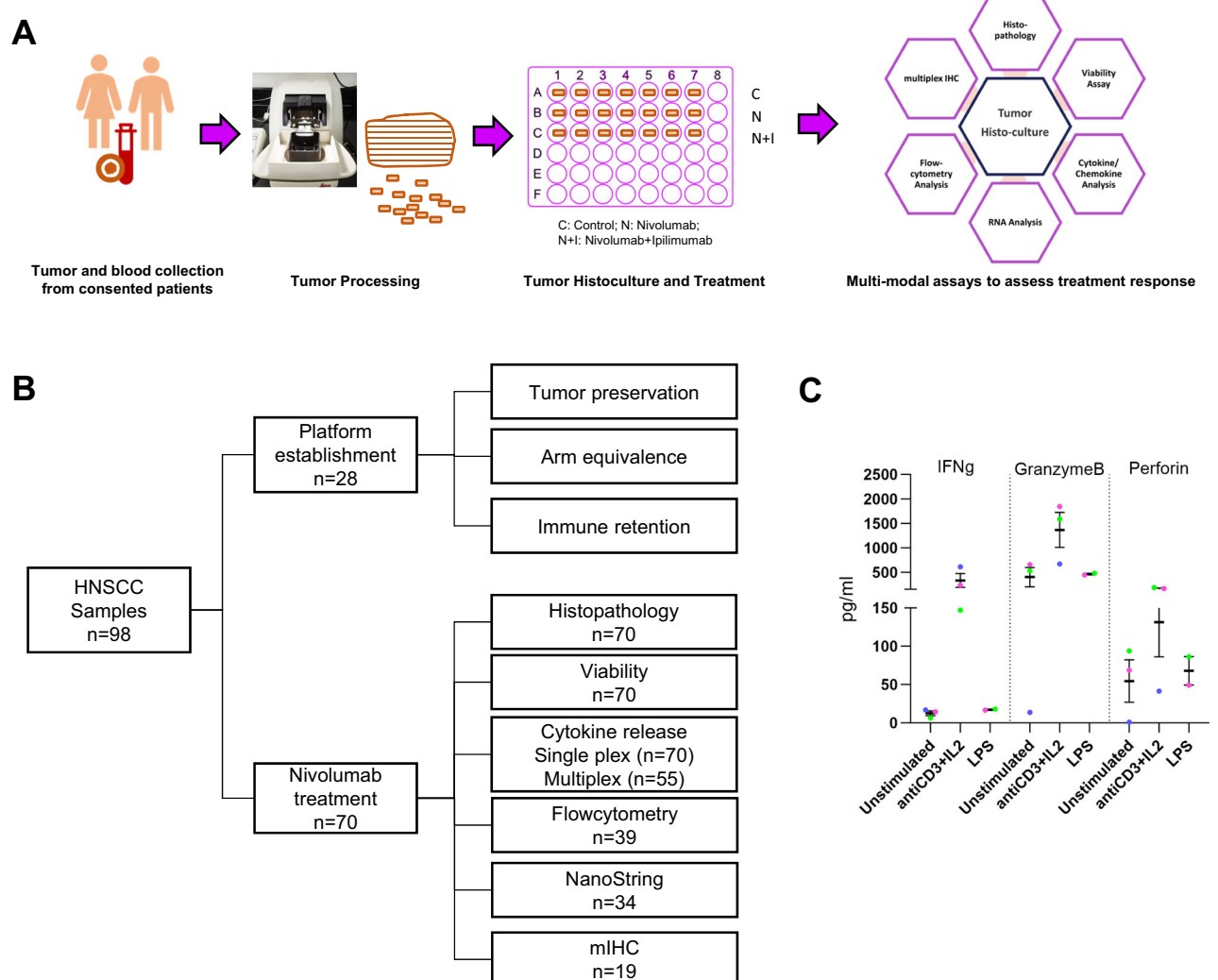

**Fig. 1 | Development of TruTumor histoculture platform to study intra-tumoral immune cell function and response to treatments. A** Fresh surgically resected HNSCC tumor samples were processed to generate thin explants and treated with agents for 72 h, and response evaluation were performed using multimodal assay readouts. **B** Sample utilization summary **C** Intra-tumoral immune cell functional fidelity was established by stimulating with antiCD3 + IL2 (*n* = 3 patient samples) or LPS (*n* = 2 patient samples) for 72 h. The response was evaluated using the secretion of IFNg, Perforin, and GranzymeB. Data represents Mean ± SEM. Each patient sample has been represented with a different color. Source data are provided as a Source Data file.

also increased CD8 + Ki67+ (*p* < 0.001) and CD8 + GZMB+ (*p* < 0.0001) sub-populations (Fig. 2C) indicative of T-cell proliferation and reinvigoration, with a significant increase in the proliferating CTL to Treg ratio (*p* < 0.001) (Fig. 2D). Increased IFNg release on treatment resulted in cohort level upregulation of multiple interferon-gamma signaling pathway genes[11] (e.g., *CXCL9*, *CXCL10*, *CXCL11*, *GBP1*, *GBP2*, *GBP4*, *IDO1*, *IRF1*, *STAT1*) (Fig. 2E).

### Sub-cohort analysis to identify true responders and non-responders

We next sought to stratify the heterogenous response to PD1 blockade on a sub-set of 55 HNSCC samples using 10 parameters associated with tumor viability and immune cell activity (detailed in Materials and methods section). Log2 fold change values for each of these parameters were used to perform an unsupervised nonlinear dimension reduction t-SNE plot to generate sample clusters. Based on the t-SNE plot, the 55 samples separated into two distinct sub-cohorts (SC) SC1 (*n* = 9) and SC2 (*n* = 46) (Fig. 3A).

In a bid to identify true non-responders, SC2 was further subdivided based on two phenotypes: T-cell reinvigoration (assessed by IFNg release) and tumor cytotoxicity (assessed by tumor content

decrease, cleaved Caspase-3 increase, GranzymeB and Perforin release). A treatment induced fold change of >1.2 compared to control for each of these parameters was considered as relevant change for that parameter. Four sub-cohorts of SC2 emerged with the following characteristics (Fig. 3B): SC2a with some degree of T-cell reinvigoration along with tumor cytotoxicity (*n* = 27); SC2b wherein tumor cytotoxicity was present but no T-cell reinvigoration was evidenced (*n* = 8); SC2c wherein T-cell reinvigoration was present without any tumor cytotoxicity (*n* = 5); SC2d wherein there was absence of both T-cell reinvigoration and tumor cytotoxicity (*n* = 6). SC1 exhibited significantly high levels of IFNg release (*p* < 0.001) compared to all other sub-cohorts. IFNg GES showed a near significant increase in SC1 in comparison to SC2d (Fig. 3C). Tumor cytotoxicity was also higher in SC1 compared to all other sub-cohorts (Fig. 3D). GranzymeB was high in SC1 in absence of treatment and did not increase further upon treatment (Supplementary Fig. 6).

Furthermore, analysis of gene expression signatures across these sub-cohorts revealed the highest expression of IFNg signaling, T-cell activity and immune cell infiltration-associated genes in SC1 compared to all other sub-cohorts. In contrast, the majority of immune suppression and tumor progression-associated genes were highly

**Table 1 | Patient demography and clinical information**

| Parameters | Categories | Values (%) |
|---|---|---|
| Age | <56 (Median) | 47 (48.0%) |
| | ≥56 (Median) | 51 (52.0%) |
| Gender | Male | 39 (39.8%) |
| | Female | 59 (60.2%) |
| Grade | Grade-1 | 68 (69.39%) |
| | Grade-2 | 26 (26.53%) |
| | Grade-3 | 2 (2.04%) |
| | Sarcomatoid variant | 2 (2.04%) |
| Stage | I | 1 (1.0%) |
| | II | 25 (25.5%) |
| | III | 33 (33.7%) |
| | IV | 32 (32.7%) |
| | NA | 7 (7.1%) |
| Sample type | Local recurrent | 3 (3.1%) |
| | Primary | 94 (95.9%) |
| | Metastatic | 1 (1.0%) |
| Neo adjuvant treatment given | Yes | 14 (14.3%) |
| | No | 76 (77.5%) |
| | NA | 8 (8.2%) |
| Tumor site | Buccal mucosa | 45 (45.9%) |
| | Tongue | 17 (17.4%) |
| | Alveolus | 10 (10.2%) |
| | Gingivobuccal sulcus (GBS) | 7 (7.1%) |
| | Lip | 5 (5.1%) |
| | Oral cavity | 9 (9.2%) |
| | Non-oral cavity | 5 (5.1%) |

upregulated in SC2d compared to other cohorts (Fig. 3E and Supplementary Table 1). Increased release of, IL1B, CXCL9, CXCL10, and GM-CSF in addition to IFNg and Perforin was observed in SC1 whereas TNF-α, MCSF, MMP-9, CXCL8, CX3CL1, CCL2 release was higher in SC2d (Fig. 3F) indicative of a CTL and M1 macrophage driven response in SC1 while a M2 and possibly Myeloid-Derived Suppressor Cells (MDSC) driven resistance phenotype in SC2d. Hierarchical clustering using both differential gene expression and cytokine release was able to clearly segregate all sub-cohorts (SC1 to SC2d). Based on these results, SC1 and SC2d were categorized as potential responder (R) and non-responder (NR) sub-cohorts respectively. Sub-cohorts SC2a-c showed varying ranges of T-cell reinvigoration and tumor cytotoxicity and were hence categorized as moderate responders.

## Comparing responders and non-responders to identify response and resistant biomarkers

Cytotoxic T-cells and the NK cells are key immune cell types which upon activation via concerted secretion of Perforin and GranzymeB causes tumor-targeted cytotoxicity[12]. We observed that activated CTLs (CD8 + /GZMB + ) were significantly higher in SC1 ($p < 0.05$) compared to SC2d even in the absence of treatment. We also observed a trend of higher NK cell activity and a near significant increase in NK cytotoxic cell gene signature in SC1 compared to SC2d upon treatment. A near significant increase in the release of Perforin/GranzymeB release upon treatment was observed in SC1 in comparison to SC2d (Fig. 4A). CTL and NK cell chemo-attractant chemokine CXCL9 and CXCL10[13] were increased in SC1 upon treatment with Nivolumab compared to SC2d[13] (Fig. 4B).

We next analyzed the spatial distribution of CD8+ CTLs in a subset of responder sub-cohort SC1 ($n = 5$) and non-responder sub-cohort SC2d ($n = 4$) using mIHC, based on tissue availability. Not only did the

responder have a higher CTL number than the non-responder in the untreated arm, these CTLs showed infiltration (evidenced by near significant decrease in distance between tumor and CTLs) into the tumor nest on Nivolumab treatment (Fig. 4C). On the contrary, the CTLs showed no significant reduction in their distance from the tumor in SC2d samples (Fig. 4C). Samples which did not show any significant decrease in CTL: tumor distance did not exhibit tumor cytotoxicity (indicated by either tumor content decrease or cleaved caspase-3 increase) (Fig. 4D). Differential gene signature analysis comparing SC1 with SC2d revealed significant upregulation of T-cell memory and Tumor Inflammation Signature (TIS)[14] and a near significant increase in Interferon-gamma gene signature in SC1 (Fig. 4E).

The immune cell subtype that dominated in SC2d's non-responsive phenotype included M2 macrophages and Tregs. SC2d had a significantly higher M2 (CD68 + CD206 + ) proportion even in the absence of any treatment (Fig. 5A). On treatment SC1 samples exhibited an increasing trend in M1/M2 GES that correlated with a near significant increase in GMCSF:MCSF ratio compared to SC2d (Fig. 5A). There was an increase in the total and proliferating Treg cells in SC2d upon treatment leading to a decrease in CTL/Treg ratio (Fig. 5B). Increased release of TNFα and IL8 was observed in SC2d upon treatment (Fig. 5C). It appeared that while the response phenotype was driven by increased immune cells that promoted anti-tumor immunity, resistance was mainly driven by the tumor cell-mediated repression of immune cell response.

## Evaluating baseline markers for correlation with response

The total CTLs in TME and those within the tumor nest were both higher in SC1 than in SC2d, though not significant (Fig. 6A). PD-L1 expression in tumor cells was evaluated using Combined Positive Score (CPS) on tumor and immune cells. Scores were between 3-100% in all samples from both cohorts. The SC1 sub-cohort contained more samples with PD-L1 CPS scores of >50 (Fig. 6B). Interestingly, SC1 samples with >50 CPS scores had a relatively higher number of CTLs within the TME. SC1 contained more Grade 1 (78%) samples compared to SC2d whereas SC2d had more Grade 2 (67%) (Fig. 6C). Tumor stage did not seem to influence response to treatment. The majority of SC1 (56%) were from buccal mucosa whereas 50% of SC2d were tongue samples. While all samples in SC2d were HPV negative, only one buccal mucosa sample which was part of the responder SC1 sub-cohort was HPV positive. HPV positivity in HNSCC is attributed to high-risk sexual behavior[15]. The low HPV positivity in this study cohort could be attributed to tobacco chewing and smoking being the primary cause for HNSCC in India[16], from where all these samples were collected.

## Baseline and on treatment gene expression signatures that predict response to treatment

Gene expression at baseline between SC1 and SC2d sub-cohorts was compared. Since alteration in signaling pathways that promote tumor progression play a role in determining response to therapy, we evaluated NanoString annotated gene expression signatures (Supplementary Table 1) using mRNA data from all samples from these two sub-cohorts. Based on the gene expression data, it was evident that the SC1 sub cohort had multiple anti-tumor immune cell activity-related pathways upregulated (Fig. 7A). Additionally, there were increased cytotoxicity and apoptosis gene expression signatures in SC1 on-treatment that supported an increase in tumor cytotoxicity as evaluated by increase in cleaved caspase-3 expression in the tumor compartment and perforin release (Fig. 3D). On the contrary SC2d had upregulation in immune suppressive (Fig. 7A) and tumor progression pathways potentially creating a more pro-tumorigenic microenvironment. TGF-beta pathway has been shown to promote immune evasion and promote resistance to anti-PD1 therapy[17]. While certain pathways, like Notch signaling[18] that were upregulated in SC2d have both pro- and anti-tumor roles, in the current context

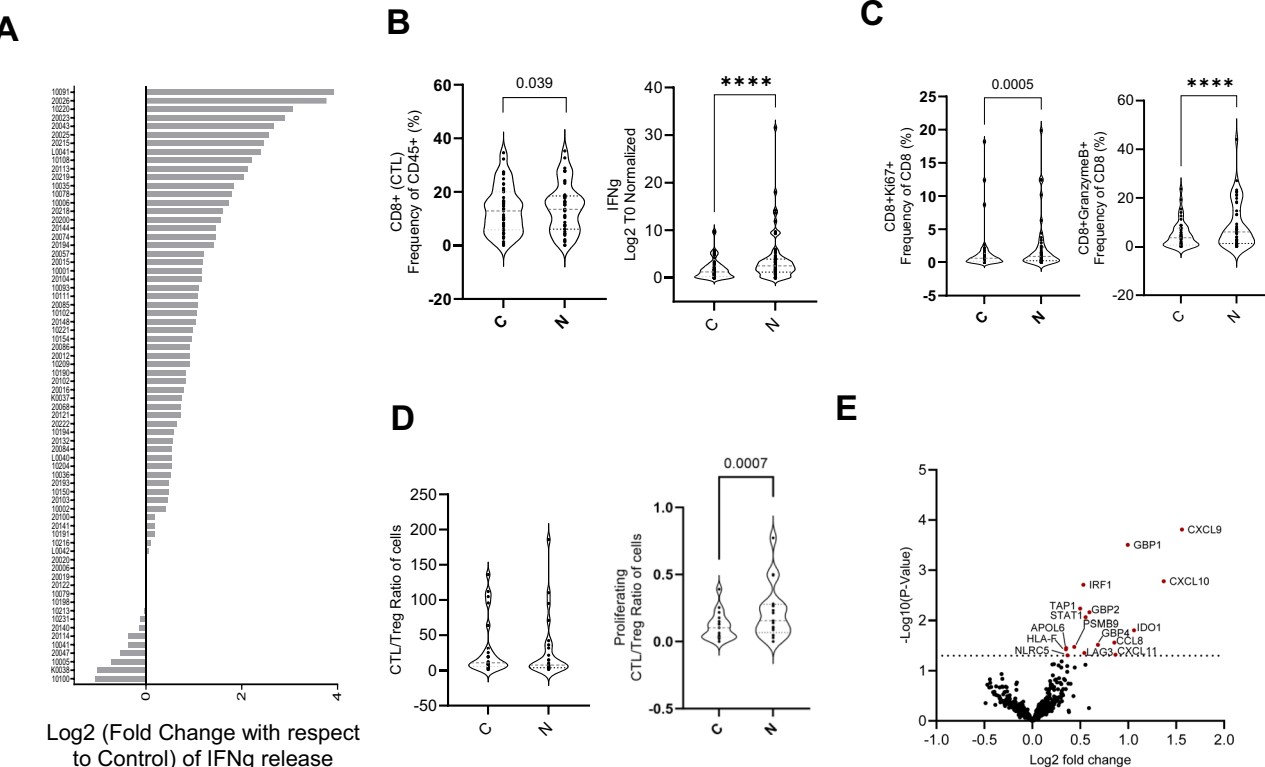

**Fig. 2 | Effect of treatment with Nivolumab on TruTumor platform at a cohort level. A** Heterogenous IFNg release response observed across patients ($n = 70$ Patient samples) shown as waterfall graph. **B** Change in CTL proportions ($n = 43$ Patient samples) and IFNg cytokine secretion ($n = 70$ Patient samples) in response to treatment. Data is represented as violin plots and p-values calculated using paired t-test **C** Change in proliferating CTL (CD8 + Ki67 + ) and activated CTL (CD8+GranzymeB + ) on treatment ($n = 43$ Patient samples). Data is represented as violin plots and p-values calculated using paired t-test **D** Effect of treatment on total ($n = 24$ Patient samples) and proliferating ($n = 17$ Patient samples) CTL/Treg ratio evaluated using flow cytometry. Data is represented as violin plots and p-values calculated using paired t-test. (All paired t-test are performed using Wilcoxon matched-pairs signed rank two tailed test, ****$p < 0.0001$) **E** Differential gene expression ($n = 34$ Patient samples) analysis upon treatment, compared to control. Highlighted in red are the genes showing a significant over-expression (Wald test *p*-value < 0.05). Source data are provided as a Source Data file.

wherein the SC2d sub-cohort exhibited a poor response to Nivolumab, the pro-tumor role of these pathways seem to be more relevant.

Of the genes with significant modulation, those with a PCA loading score of more than 0.7 (Supplementary Fig. 7C) were selected. Of these, 12 were found to overlap with a clinical dataset ($n = 102$) from HNSCC patients that were treated with anti-PD1 therapy[19]. All except one responder sample clustered together based on the expression pattern of these 12 genes (Supplementary Fig. 7D). Gene expression scores computed from these 12 genes exhibited a near significant segregation of the responder (SC1) and non-responder (SC2d) samples with an AUC of 0.88 (p-value = 0.02) from TruTumor ROC curve (Supplementary Fig. 7E). Scores from all other sub-cohorts (SC2a, SD2b and SC2c) were in between SC1 and SC2d. Significant segregation was observed between responders (including Complete and Partial Responders, CR/PR) and non-responder samples (Progressive Disease, PD) with those with Stable Disease (SD) falling in between. Based on a Youden cut-off score of 6.28 derived from clinical data ROC curve (Supplementary Fig. 7E), there was a significant separation ($p = 0.0012$) in Progression Free Survival (PFS) (Fig. 7C) with a higher score corresponding to improved survival.

On analysis of on-treatment data (detailed in materials and methods), the expression of seventy-eight genes were found to be significantly altered in the treated arm of SC1 compared to SC2d (Supplementary Fig. 8B). These genes were able to segregate all except two samples from SC1 and SC2d by PCA analysis into well-separated clusters (Supplementary Fig. 8C). Of the 78 genes, 31 were found to overlap with a metastatic Melanoma clinical dataset published by Chen et al.[20]. The GES score was significantly higher in the responder

samples in both the TruTumor (AUC of 0.89, p-value 0.01 derived from TruTumor ROC curve, Supplementary Fig. 8E) and clinical datasets (Fig. 7D). The GES scores of sub-cohorts 2a-c were in between those of SC1 and SC2d. The 31 genes were able to segregate the SC1 and SC2d samples into distinct clusters by hierarchical clustering except for one SC1 sample (Supplementary Fig. 8D). The genes of the On-treatment GES belonged to pathways related to immune cell migration and function (T-cell, NK cells, T effector memory, macrophages). In addition, NF-κb and immune-suppression-related genes were also present indicating the onset of activity-related exhaustion of immune cells.

A cumulative analysis of all parameters that significantly correlated with higher baseline and on-treatment GES scores revealed that the on-treatment scores exhibited a significant correlation with T cell and NK cell activity, tumor cytotoxicity, and T cell honing into tumor compared to baseline scores. This indicates that the responders are pre-primed to respond to anti-PD1 therapy at baseline. Up on treatment these response parameters improved significantly (Fig. 7E).

### Combination with Ipilimumab confers improved efficacy in moderate responders

SC2a constituted the majority (68%) of all moderate responders. While this sub-cohort exhibited a significant increase in IFNg release post Nivolumab treatment (Fig. 3C), tumor cytotoxicity was not as high as in SC1 (Fig. 3D). We assessed if Ipilimumab combination treatment would improve tumor cytotoxicity in SC2a.

The inclusion of Ipilimumab reduced IL-10 secretion with concomitant Treg depletion (Fig. 8A). A significant increase in CTL/Treg ratio was observed (Fig. 8A right panel) on treatment with the

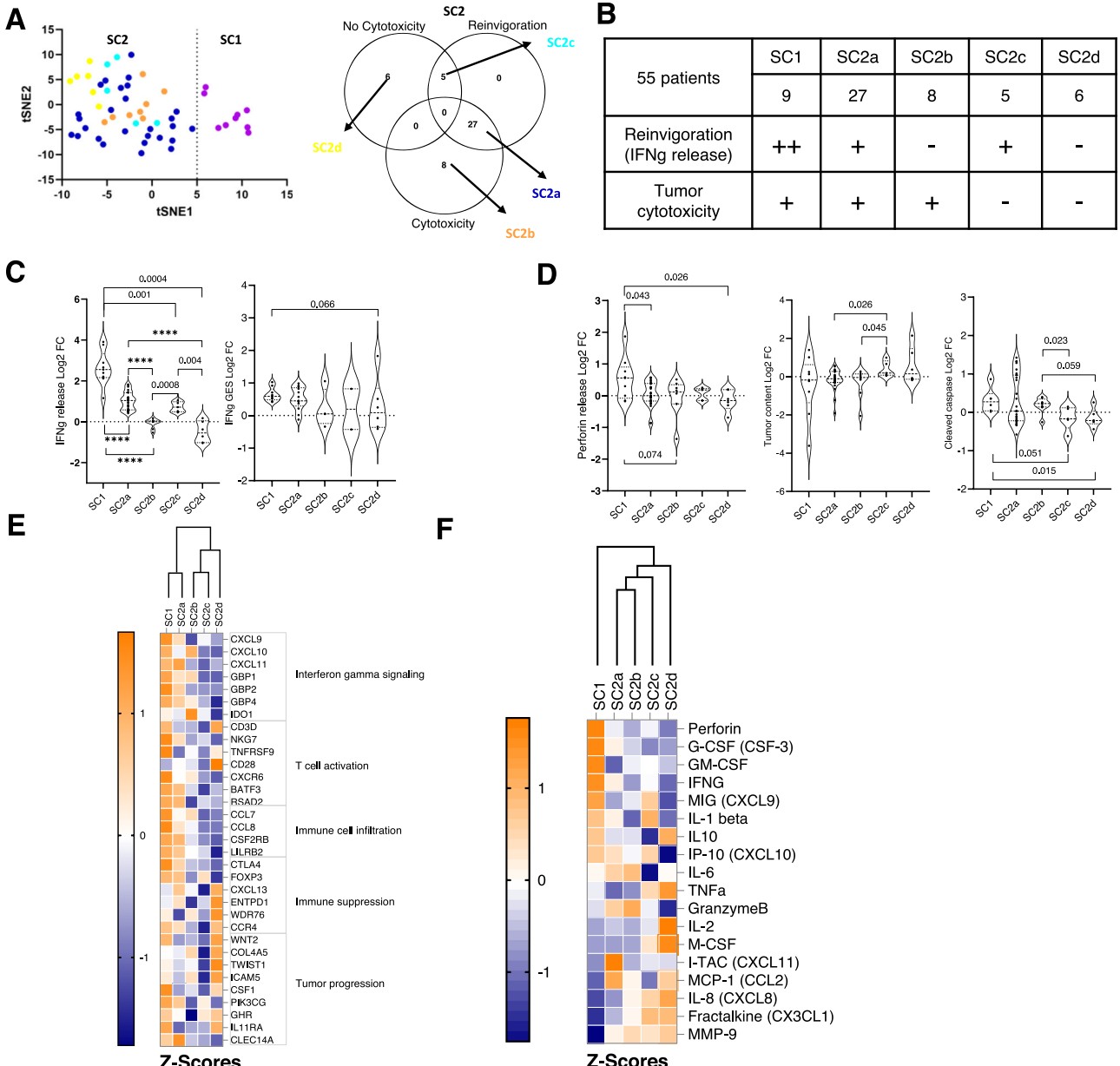

**Fig. 3 | Sub-cohort stratification based on tSNE generated using 10 response parameters. A** Identification of potential responder sub-cohort (SC1: *n* = 9 Patient samples, Purple) and moderate/poor responders (SC2: *n* = 46 Patient samples, Left panel). Sub-stratification of SC2 into SC2a-d (SC2a: Dark blue, SC2b: Orange, SC2c: Light blue and SC2d: Yellow) based on overlap between T-cell reinvigoration (Reinvigoration) measured by IFNg release and cytotoxicity or no cytotoxicity phenotype measured by decrease in tumor content or increased tumor cell cleaved caspase-3 expression or increase of Perforin or GranzymeB release. **B** Table for all the 6 sub-cohorts denoting the number of samples and their properties. **C** Variation in IFNg release (Left, SC1: *n* = 9 Patient samples, SC2a: *n* = 27 Patient samples, SC2b:*n* = 8 Patient samples, SC2c: *n* = 5 Patient samples and SC2d: *n* = 6 Patient samples) and IFNg gene expression signature (right, SC1: *n* = 9 Patient samples, SC2a: *n* = 13 Patient samples, SC2b:*n* = 4 Patient samples, SC2c: *n* = 2 Patient samples and SC2d: *n* = 6 Patient samples) across sub-cohorts. Data is represented as violin plots and p-values calculated using un-paired t-test. **D** Post treatment variation in cytotoxicity parameters assessed by increase in perforin release (SC1: *n* = 9 Patient samples, SC2a: *n* = 27 Patient samples, SC2b: *n* = 8 Patient

samples, SC2c: *n* = 5 Patient samples and SC2d: *n* = 6 Patient samples), decrease in tumor content (SC1: *n* = 9 Patient samples, SC2a: *n* = 27 Patient samples, SC2b:*n* = 8 Patient samples, SC2c: *n* = 5 Patient samples and SC2d: *n* = 6 Patient samples), and increase in the expression of cleaved caspase-3 (SC1: *n* = 6 Patient samples, SC2a: *n* = 27 Patient samples, SC2b:*n* = 8 Patient samples, SC2c: *n* = 5 Patient samples and SC2d: *n* = 6 Patient samples) across sub-cohorts. Data is represented as violin plots and p-values calculated using un-paired t-test. (All un-paired t-test are performed using non-parametric Mann Whitney two tailed test, numerical value for ****$p$ < 0.0001) **E** Heatmap of Log2 gene expression fold change across sub-cohorts with respect to their respective control determined by differential gene expression analysis represented as Z-scores. Annotations of pathways based on literature is provided in Supplementary File SF1. The columns have been grouped using Euclidean distance based unsupervised clustering. **F** Heatmap of fold change in cytokines release with respect to control. Data is represented as Z-scores. The columns have been grouped using Euclidean distance based unsupervised clustering. Source data are provided as a Source Data file.

combination. The ratio seemed to be driven mainly by Treg depletion rather than improved T-cell function as evident from the absence of any significant change in IFNg release (Fig. 8B left). The suppression of Treg, however, resulted in improved tumor cytotoxicity (Fig. 8B right).

The tumor cytotoxicity in SC2a as measured by tumor-cleaved Caspase 3 expression on combination treatment matched the SC1 response to Nivolumab monotherapy (Fig. 8B). To investigate response at a sample level, nine samples with varying levels of improved tumor cytotoxicity

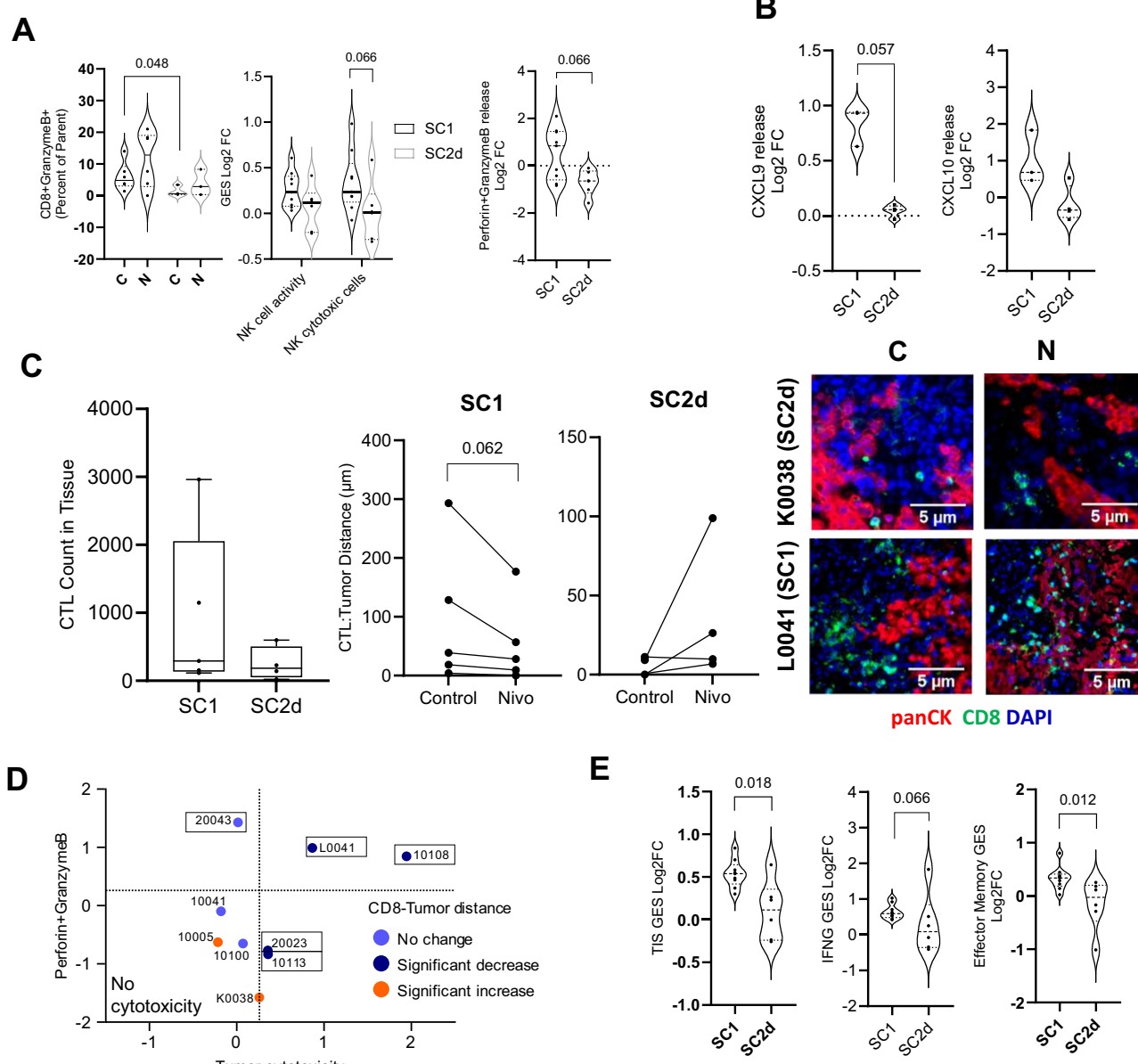

**Fig. 4 | Characteristics of potential responders. A** Proportions of activated CTL (CD8 + /GranzymeB + ) assessed by flow cytometry in the responder (SC1, $n = 6$ Patient samples) and non-responder (SC2d, $n = 3$ Patient samples) sub-cohorts. Fold change with respect to control in NK cell function (assessed by NanoString based GES) and tumor cytotoxicity inducing proteins (Perforin + GranzymeB) release in the two sub-cohorts. Data is represented as violin plots and p-values calculated using un-paired t-test. **B** CXCL9 and CXCL10 cytokine release fold change compared to control, up on treatment with Nivolumab in SC1 ($n = 3$ Patient samples) and SC2d ($n = 4$ Patient samples) sub-cohorts. Data is represented as violin plots and p-values calculated using un-paired t-test. **C** The left panel shows the image analysis data representing the total CTLs (CD8+ cells) in the TME from $n = 5$ Patient samples of SC1 and $n = 4$ Patient samples of SC2d. Data is represented as box plot, where the middle line represents the median and the lower and upper hinges represent the 25th and 75th percentile respectively, the whiskers extend from the hinge to the minimal and maximal data point but no further than 1.5 times interquartile range. The median distance between the CTLs and panCK+ tumor region upon treatment in SC1 and SC2d are represented as symbols and line plot

(p-values calculated using paired t-test). The right panel shows a representative multiplexed IHC (mIHC) image displaying the relative spatial distribution of CTLs (CD8 + ; green) and tumor cells (panCK, red) in a SC1 sample (L0041) and SC2d sample (K0038). DAPI (blue) was used to stain nuclei. **D** The two-dimensional plot represents the cytotoxicity parameters as Log2 fold change with respect to control ($n = 9$ Patient samples). x-axis represents the tumor cytotoxicity (negative of tumor content decrease or increased caspase expression which shows greater cytotoxicity and y-axis Perforin and GranzymeB release). Significant decrease (Deep blue), no change (Light Blue), and significant increase (Orange) in CD8: tumor distance are represented. The SC1 samples are demarcated by boxes. **E** The fold change compared to control for GES that were significantly different between the two sub-cohorts (SC1: $n = 9$ Patient samples and SC2d: $n = 6$ Patient samples). Data is represented as violin plots and the p-value is calculated using unpaired t-test. All unpaired t-test were performed using non-parametric Mann Whitney two tailed test and paired t-test were performed using Wilcoxon matched-pairs signed rank two-tailed test. Source data are provided as a Source Data file.

on combination treatment were selected for spatial analysis (Fig. 8C). Using a mIHC approach, Treg numbers were found to have a decreasing trend on combination treatment (Fig. 8C, left). Distance between CTL and Treg significantly increased in samples that exhibited

enhanced tumor cytotoxicity (tumor content decrease or caspase increase) on combination treatment while showing no change or decrease in samples where enhanced cytotoxicity was not observed (Fig. 8C, middle). These data suggest the importance of both the

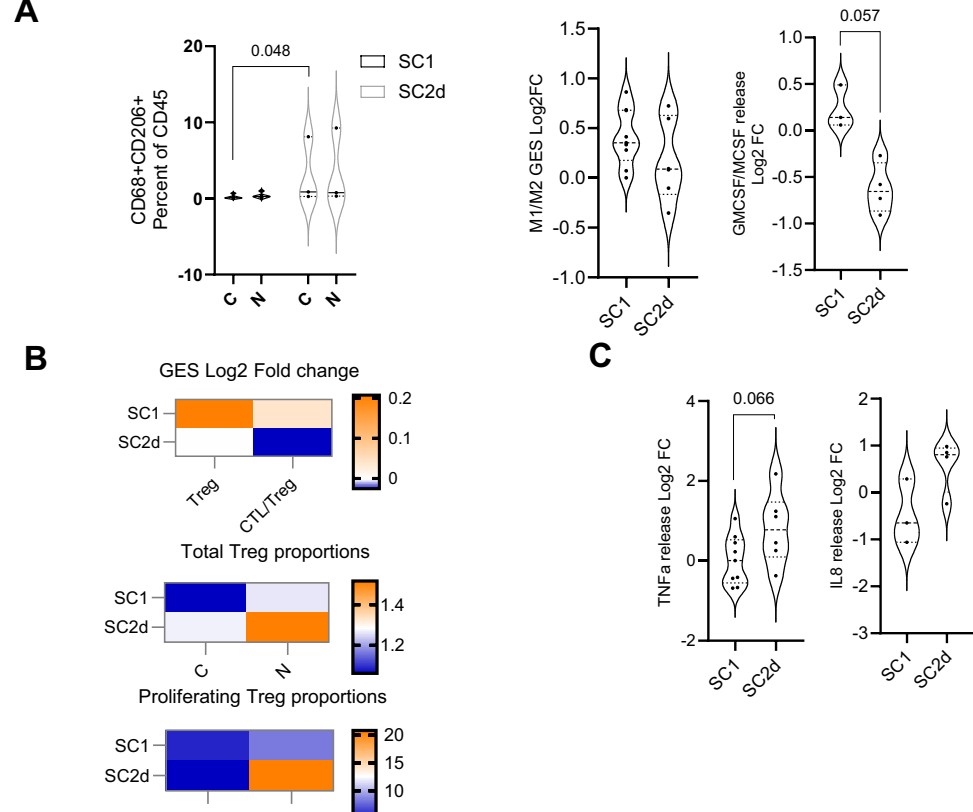

**Fig. 5 | Characteristics of non-responders or treatment resistant sub-cohort.**
**A** Multiple assay readouts indicative of an M2 macrophage polarization are represented. Left panel depicts Flow cytometry-based evaluation of CD68 + CD206 + M2 like macrophages in SC1 ($n$ = 6 Patient samples) and SC2d ($n$ = 3 Patient samples). Middle panel depicts ratio of M1/M2 using NanoString derived GES (SC1, $n$ = 9 Patient samples and SC2d, $n$ = 6 Patient samples) expressed as Log2 fold change compared to control. Right panel depicts GMCSF/MCF release expressed as Log2 fold change compared to control for each sample of SC1 ($n$ = 3 Patient samples) and SC2d ($n$ = 4 Patient samples). Data is represented as violin plots and p-values calculated using un-paired t-test. **B** Topmost panel is the heatmap representation of effect of Nivolumab treatment on Treg and CTL/Treg GES represented

as Log2 fold change with respect to control (SC1, $n$ = 9 Patient samples and SC2d, $n$ = 6 Patient samples). The middle and bottom panels depict proportion of total Treg (CD4+FoxP3+ as % of CD45) and proliferating Tregs (CD4+FoxP3+ Ki67+ as % of CD4+FoxP3+ ) determined by flow cytometry in control and Nivolumab treated arms in SC1 ($n$ = 6 Patient samples) and SC2d ($n$ = 3 Patient samples). Data represents mean value of the sub-cohorts. **C** Release of TNF-a (SC1, $n$ = 9 Patient samples and SC2d, $n$ = 6 Patient samples) and IL8 (SC1, $n$ = 3 Patient samples) and SC2d ($n$ = 4 Patient samples) expressed as Log2 fold change compared to control, are represented graphically. Data is represented as violin plots and p-values calculated using un-paired t-test. (All un-paired t-test are performed using non-parametric Mann Whitney two tailed test). Source data are provided as a Source Data file.

---

number of Treg and their physical proximity with CTL in driving response to Nivolumab and Ipilimumab combination.

## Discussion

Anti-PD1 therapy provides a durable response against multiple solid tumors. However, with a response rate of only about 18% in PD-L1 positive HNSCC patients, it remains a challenge to identify responder patients[21]. Tumor mutation burden (TMB), an emerging predictive marker for ICI response potentially through enhanced tumor antigen presentation, also suffers from high false positivity rates[22] like PD-L1 score, highlighting the need for better predictive markers.

Multiple factors pertaining to the tumor micro-environment determine the efficacy of anti-PD1 response. Immunoediting mechanisms adopted by tumors contribute to resistance to anti-PD1 therapy. Resistance is contributed by one or many of the following factors such as irreversible T-cell exhaustion, dysfunctional antigen presentation, resistance to IFNg signaling, and tumor/stroma-driven immunosuppressive environment[23]. The inadequacy of a single biomarker to reliably predict response is explained by the wide range of resistance mechanisms listed above that are driven by the complex and evolving micro-dynamics of the tumor micro-environment.

Tumor histoculture provides the most comprehensive representation of TME. The TruTumor histoculture platform exhibited anti-

PD1 treatment response phenotypes similar to other histoculture platforms reported in the literature (Supplementary Table 2). The ability of the TruTumor platform in effectively combining bulk with spatial data from multiple different assay read-outs makes it unique compared to all other reported platforms.

The response rate of the TruTumor platform to Nivolumab treatment was ~16% (9/55, Fig. 3B) closely mimicking the clinical response rate. Samples in both SC1 and SC2d from the current study were predominantly HPV negative and positive for PD-L1 expression. The PD-L1 scores ranged from 4-100% in the non-responder (SC2d) cohort indicating poor response predictive power of PD-L1. No correlation was observed between PD-L1 score and level of response to Nivolumab treatment.

Anti-PD1 therapy driven increase in tumor specific T-cell repertoire[24] implicates the role of recruitment of peripheral T-cells into the tumor upon treatment. This aspect of anti-PD1 response is not possible to study in any ex vivo platform. However, a significant increase in intra-tumoral CD8 + T-cells (CTLs) that was observed at a cohort level post-treatment in the current study (Fig. 2A), along with a highly significant increase in IFNg secretion and up-regulation of IFNg pathway genes (Fig. 2A, E) alleviates the need for a functional circulatory system for demonstrating response. A significant treatment-induced increase in both proliferating and activated T-cells (Fig. 2C),

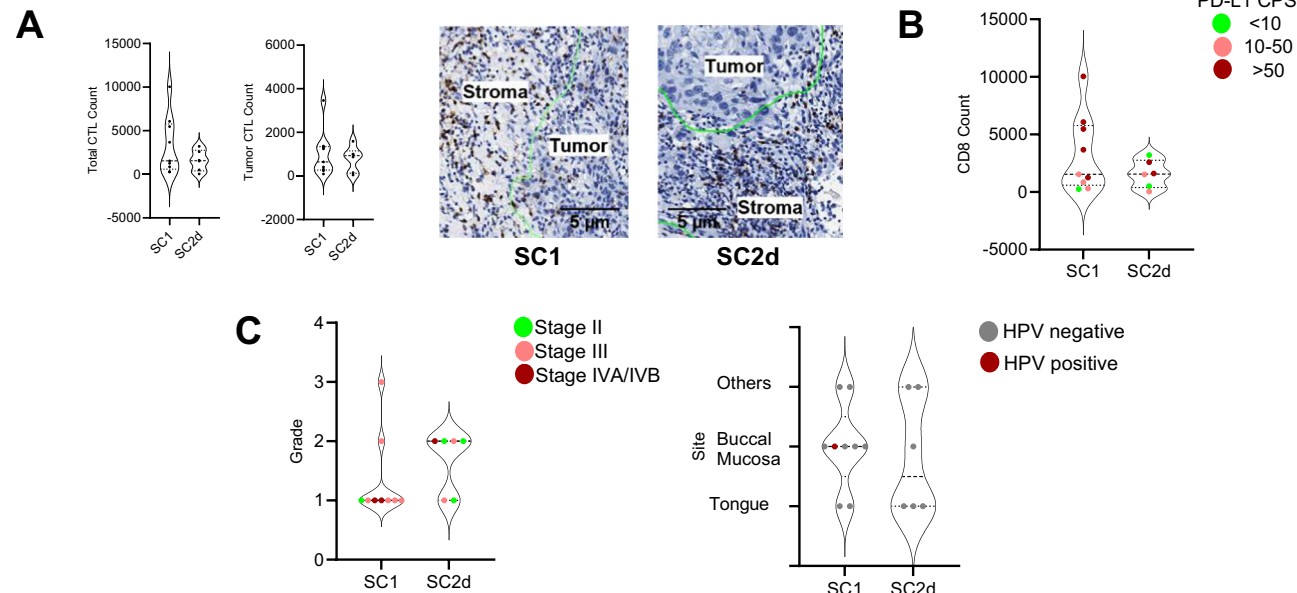

**Fig. 6 | Baseline characteristics of responder (SC1, *n* = 9 Patient samples) and non-responder (SC2d, *n* = 6 Patient samples) sub-cohorts. A** CD8+ CTLs were scored using CD8 stained baseline sample IHC slides from SC1 and SC2d. Data is represented as violin plots. Total CTLs (left) and CTLs within the tumor nest (middle) are graphically represented. The right panel depicts IHC images of CD8 stained slides from SC1 and SC2d sub-cohorts. **B** Correlation between CTL numbers with PD-L1 CPS scores is represented for the SC1 and SC2d sub-cohorts. PD-L1 scores, divided into three buckets (< 10, 10–50, and >50), are shown for each sample. **C** Correlation of clinical parameters like grade and stage with response levels is shown in the left panel. The right panel is the representation of the correlation between tumor sites and HPV positivity with response to Nivolumab treatment. Source data are provided as a Source Data file.

with a concomitant increase in the release of T-cell chemoattractant CXCL9 and CXCL10 (Fig. 4B), indicated a strong T-cell reinvigoration response. The T-cell reinvigoration response was, however, not sufficient to drive tumor cytotoxicity across all samples. While we could identify potential responders (SC1) and non-responders (SC2d), 75% of samples belonged to sub-cohorts (SC2a-c) that were moderate responders (Fig. 3B) highlighting the need for combinatorial treatment regimens to improve outcomes for the moderate responders.

The spatial contexture of pro and anti-tumor immune cells plays a critical role in determining the efficacy of anti-PD1 therapy[25,26]. The absence of tumor killing even in hot tumors with infiltrated CTLs could be better explained by the lack of spatial re-distribution of CTLs post-treatment in these samples. The CTL infiltration into the tumor nest was higher in the responder sub-cohort (SC1) than in the non-responders (SC2d) (Fig. 4C). On combination treatment with anti-CTLA4 (Ipilimumab), besides depletion of Treg and lowering of IL10 release, the spatial dynamics between CTL and Treg affected the incremental efficacy in tumor killing on combination treatment correlated with significant increase in distance between CTL and Treg (Fig. 8C).

Tumor intrinsic factors emerged as a determining force driving poor response to anti-PD1 therapy in the current study (Fig. 7A). Notch, MAPK and TGF-beta signaling were upregulated both at baseline and on-treatment in SC2d (Fig.7A), indicative of progressive disease. Wnt ligand signaling which is implicated in anti-PD1 resistance[27] was higher in SC2d compared to SC1 at baseline. Aberrant *Wnt* signaling has been shown to maintain the stemness of cancer stem cells in multiple solid tumor indications, including HNSCC[28–32]. T cell activity and antigen presentation-related pathways were appreciably higher in SC1 on treatment compared to baseline (Fig. 7A).

Response to treatment depends on how effectively the immune response overtakes the tumor response to achieve tumor cytotoxicity (Fig. 9). Combination treatment strategies have been shown to improve efficacy of anti-PD1-based treatment[33]. Treatment outcome improves on combining anti-PD1 with a targeted therapy by enhancing tumor death-induced T cell infiltration and/or by targeting tumor-induced immunosuppressive pathways to counter resistance-inducing mechanisms[34]. Additionally, resistance prediction markers are significantly worse in pre-treatment or baseline samples than in post-treatment samples[20]. Gene expression signatures involving T cell activation are significantly upregulated in responders of early on-treatment samples and not as robust in pre-treatment samples[20]. These observations are similar to response patterns observed in the TruTumor platform without the need of invasive biopsies. TruTumor platform is able to discern early on-treatment response signatures, stratify patients into poor, moderate, and good responders prior to actual treatment, and make informed decisions on the optimal treatment regimens for patients. In case of a moderate or poor response to Nivolumab treatment, we propose either a combination treatment with Ipilimumab or a combination with targeted therapy selected based on the response observed in culture could enhance the efficacy of treatment.

The TruTumor platform combines the ability to study these micro-dynamics at gene and protein levels, along with the spatial contexture of tumor and immune cells. The ability of baseline and on-treatment GES scores in stratifying clinical responders and non-responders from unrelated clinical datasets is an encouraging first step towards developing a clinically validated personalized treatment test using this platform. We were however limited by the number of genes that overlapped between the panel used in this study and the clinical datasets. This could have resulted in omission of additional genes that might have strengthened the predictive power of the gene expression signatures. A matched patient observational trial, that is currently ongoing, would further establish the clinical validity of these gene expression biosignatures.

## Materials

### Donor sample recruitment

Donor tissue specimens along with matched blood samples were obtained from consented patients diagnosed with Head and Neck Squamous Cell Carcinoma (HNSCC). Individual Institutional Ethics Committee (IEC) instituted by each sample collection center, namely

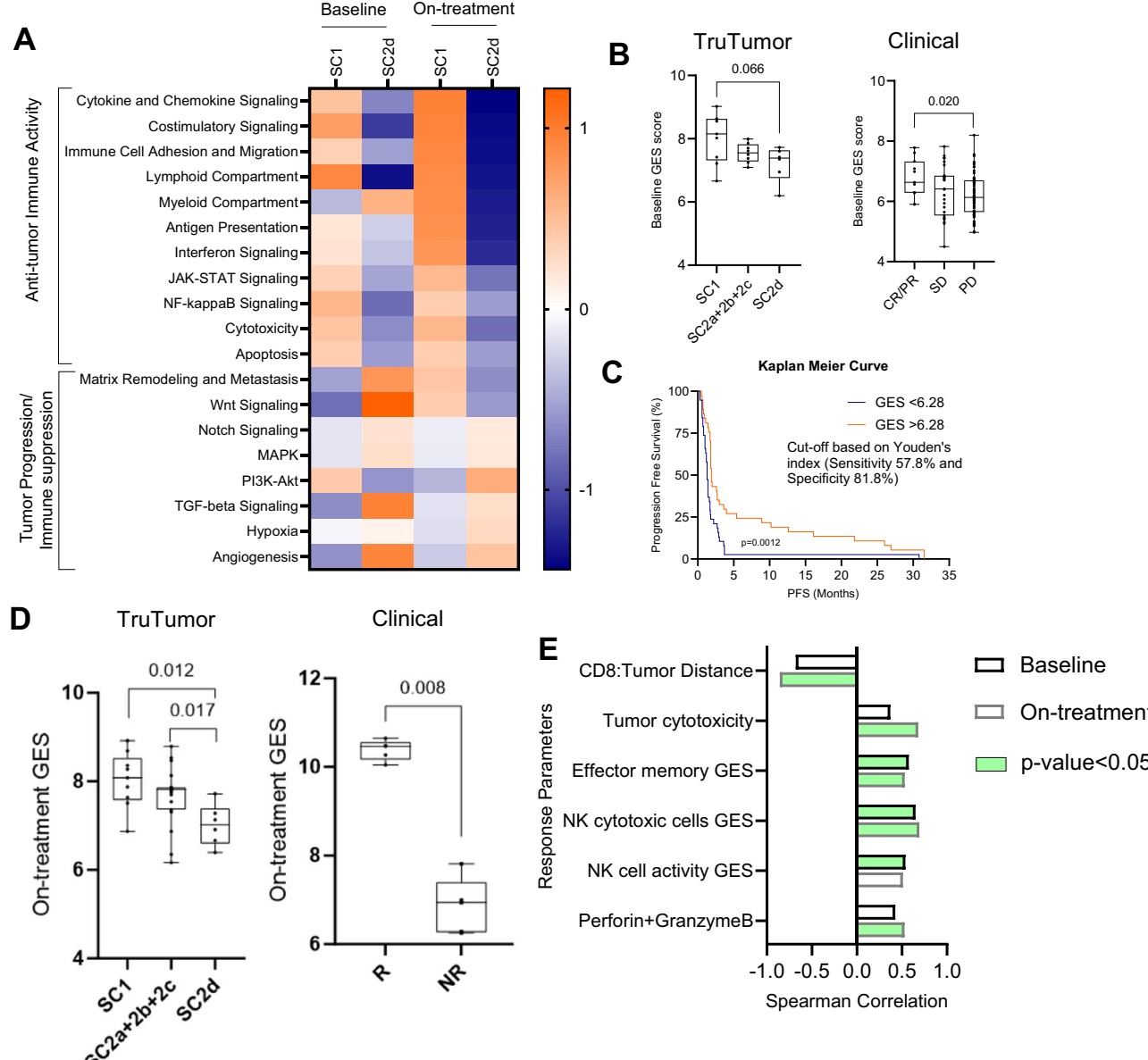

**Fig. 7 | Gene expression signatures that predict response to treatment.**
**A** Heatmap representation of pathway scores of SC1 (*n* = 9 Patient samples) and SC2d (*n* = 6 Patient samples) at baseline and on-treatment for immune activity, tumor cytotoxic and tumor activity related pathways. **B** Box plot representation of baseline GES for SC1 (*n* = 9 Patient samples), SC2a-c (*n* = 8 Patient samples), and SC2d (*n* = 6 Patient samples) shown on the left. Box plot representation of baseline GES for Responders (CR/PR), Stable Disease (SD), and non-responders (PD) from the clinical dataset (Foy et al.)[19] is shown on the right. Significance was determined using unpaired t-test. **C** Kaplan-Meier curve depicting progression-free survival was plotted for the clinical dataset, *n* = 75 Patient samples (Foy et al.)[39] using Youden's GES cut-off of 6.28 and the p-value was determined using Log-rank (Mantel-Cox) test. **D** Box plot representation of On-treatment GES for SC1 (*n* = 9 Patient samples), SC2a-c (*n* = 19 Patient samples), and SC2d (*n* = 6 Patient samples) shown on the left.

Box plot representation of on-treatment GES for responders and non-responders from the clinical dataset (Chen et al.)[20] is shown on the right. Significance was determined using an unpaired t-test. (All unpaired t-tests are performed using a non-parametric Mann-Whitney two-tailed test.) **E** Non-parametric Spearman correlation between key response parameters at baseline and on-treatment is graphically represented. GES along with their two-tailed test p-value significance is graphically represented. Parameters showing significant correlation are marked in green. For figures **B** and **D**, data is represented as a box plot, where the middle line represents the median and the lower and upper hinges represent the 25th and 75th percentile respectively, the whiskers extend from the hinge to the minimal and maximal data point but no further than 1.5 times interquartile range. Source data are provided as a Source Data file.

Vydehi Institute of Medical Sciences & Research Centre, Bangalore, Karnataka, Sri Lakshmi Multi-Specialty Hospital, Bangalore, Karnataka, Mazumdar Shaw Medical Center, Bangalore, Karnataka, Bangalore Baptist Hospital, Bangalore, Karnataka, DBR & SK Super Specialty Hospital, Tirupati, Andhra Pradesh, India, approved the protocol (protocol # FCB-PROTOCOL-01). Informed consent for participation in the approved study was obtained from every donor. Age and gender information of patients were provided by the collection centers after

redacting other personal patient identifiers. Tissue was transported in a tissue transport medium along with a matched blood sample at 4⁰C to the lab, immediately after surgery. Patients infected with HBV, HIV, and COVID-19 were excluded from the study.

**TruTumor histoculture platform**
HNSCC tissues were processed to generate thin explants that were distributed into arms containing 7 explants. One arm was submitted

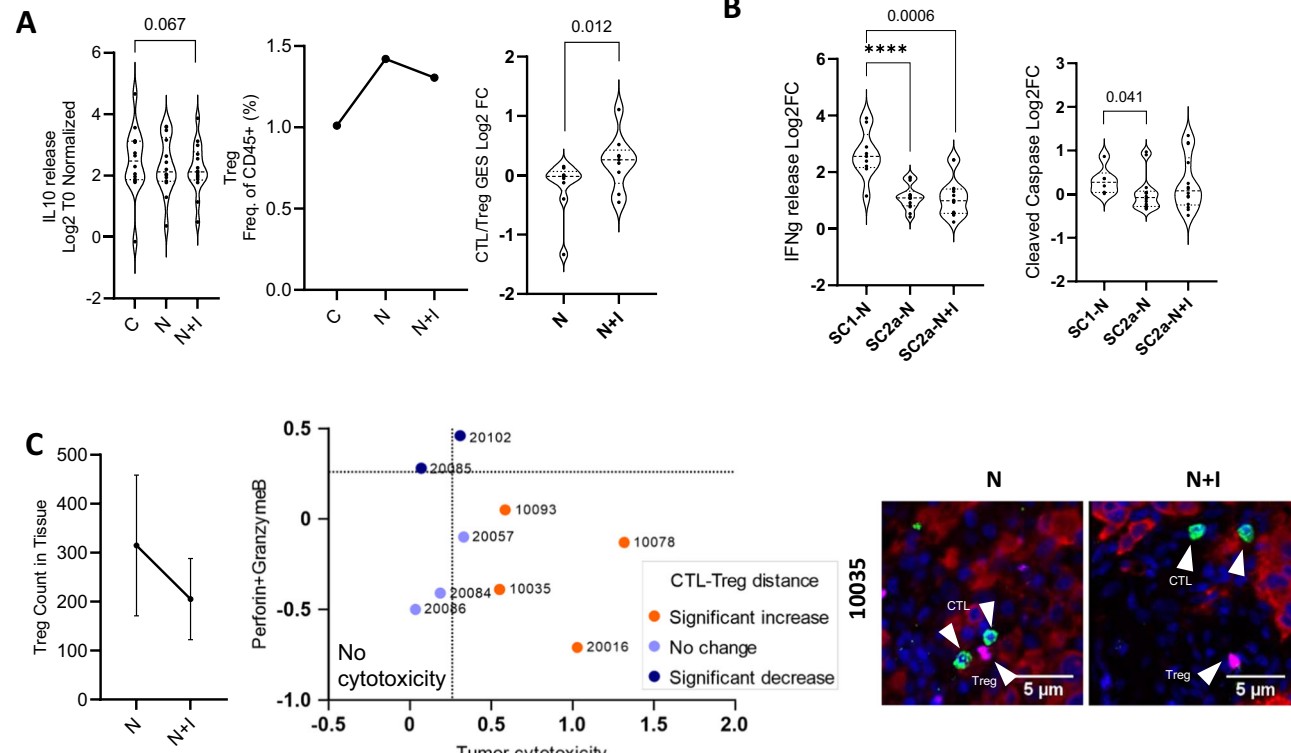

**Fig. 8 | Response of sub-cohort SC2a to Nivolumab combination with Ipilimumab. A** Left panel represents the effect of treatment on IL-10 cytokine release (*n* = 14 Patient samples). The middle panel represents a flow cytometry-based evaluation of the depletion of Tregs (*n* = 11 Patient samples). The right panel depicts the increase in CTL/Treg ratio evaluated by NanoString gene expression signature upon treatment (*n* = 9 Patient samples). Data is represented as violin plots and p-values calculated using paired t-test. **B** Comparison of response to combination treatment and Nivolumab alone with respect to Interferon gamma release and cleaved caspase expression in the tumor compartment is represented in the left and right panels respectively (SC1-N: *n* = 9 Patient samples, SC2a-N: *n* = 14 Patient samples, SC2a-N + I: *n* = 14 Patient samples). Data is represented as violin plots of fold change compared to control and p-values calculated using unpaired t-test. (All unpaired t-test were performed using a non-parametric Mann-Whitney two-tailed test and paired t-test were performed using Wilcoxon matched-pairs signed rank two-tailed test.) **C** Correlation between tumor cytotoxicity and CTL-Treg distance change on treatment with *n* + I combination with respect to N monotherapy (*n* = 9 Patient samples) Data represents Mean ± SEM. mIHC was performed using panCK for the tumor (red), CD8 for CTLs (green), FoxP3 for Tregs (purple), and DAPI for the nucleus. Treg count was compared across both treatments (left panel). The middle panel shows a two-dimensional plot representing the cytotoxicity parameters as Log2 fold change with respect to N. The x-axis represents the tumor cytotoxicity (represented as a higher value among the negative of tumor content decrease or increased caspase expression). the y-axis represents Perforin + GranzymeB secretion fold change in N + I arm with respect to N monotherapy. Distance between CTLs and Tregs measured for N and N + I combination are represented as a significant increase (orange), no change (light blue), and a significant decrease (deep blue). The representative image on the right. Source data are provided as a Source Data file.

for pre-culture histopathological analysis. The remaining arms were cultured in a culture medium supplemented with autologous plasma and FBS. The explants were treated with immune stimulants LPS (1 μg/ml), anti-CD3 (10 ng/ml) plus IL2 (100 units/ml) or treated with either with IgG4 (Control, C) or with Nivolumab (132 μg/ml, N) or with Nivolumab (132 μg/ml) and Ipilimumab (90.8 μg/ml) ( N + I) and cultured for 72 h. PrestoBlue assay was performed pre and post-culture. Culture supernatant was collected at 24 h intervals ($T_{24}$, $T_{48}$ and $T_{72}$) starting from pre-treatment ($T_0$) followed by replenishment with fresh media containing treatment agents. The collected supernatants were analyzed for cytokine release. At the end of culture, tumor explants were either fixed in 10% neutral buffered formalin followed by generation of Formalin Fixed Paraffin Embedded (FFPE) blocks or dissociated for performing flowcytometry analysis.

### Viability assay
PrestoBlue assay was carried out using PrestoBlue HS reagent (Thermo Scientific). Briefly, cell culture supernatant was replaced with PrestoBlue reagent diluted in media in each well and incubated at 37 °C with 5% $CO_2$ for 1 hr. Media blanks were included for background subtraction. At the end of incubation, supernatant

from each well was transferred into 96-well plate and the fluorescence was read using BioTek Plate Reader as per the manufacturer's instructions.

### Flow cytometry analysis
Single cells were prepared from tumor explants as per MACS Miltenyi tumor dissociation protocol using three enzyme combinations, including Enzyme H, Enzyme R, and Enzyme A. Cells were stained with Live/Dead fixable stain followed by Fc blocking. Surface marker staining with antibody cocktail containing anti-CD45, anti-CD3, anti-CD8, anti-CD4, anti-PD1 (BD Bioscience), anti-CTLA4, anti-CD56, anti-CD206, anti-CD15, anti-CD14 (Biolegend) was done for 1 hr at 4 °C. At the end of incubation, cells were fixed and permeabilized using fixation and permeabilization buffer. Intracellular marker staining with antibody cocktail containing anti-FoxP3, anti-Ki67 (BD Biosciences), anti-CD68, anti-GranzymeB (Biolegend), and anti-panCK (Novus Biologicals) was added to the cell suspension and incubated for 1 h at 4 °C. The data was acquired using a BD LSR Fortessa flow cytometer. The flow data was analyzed using FlowJo software (Version 10.8). Antibody details are provided in supplementary table 3. The gating strategy is provided in Supplementary Fig. 9.

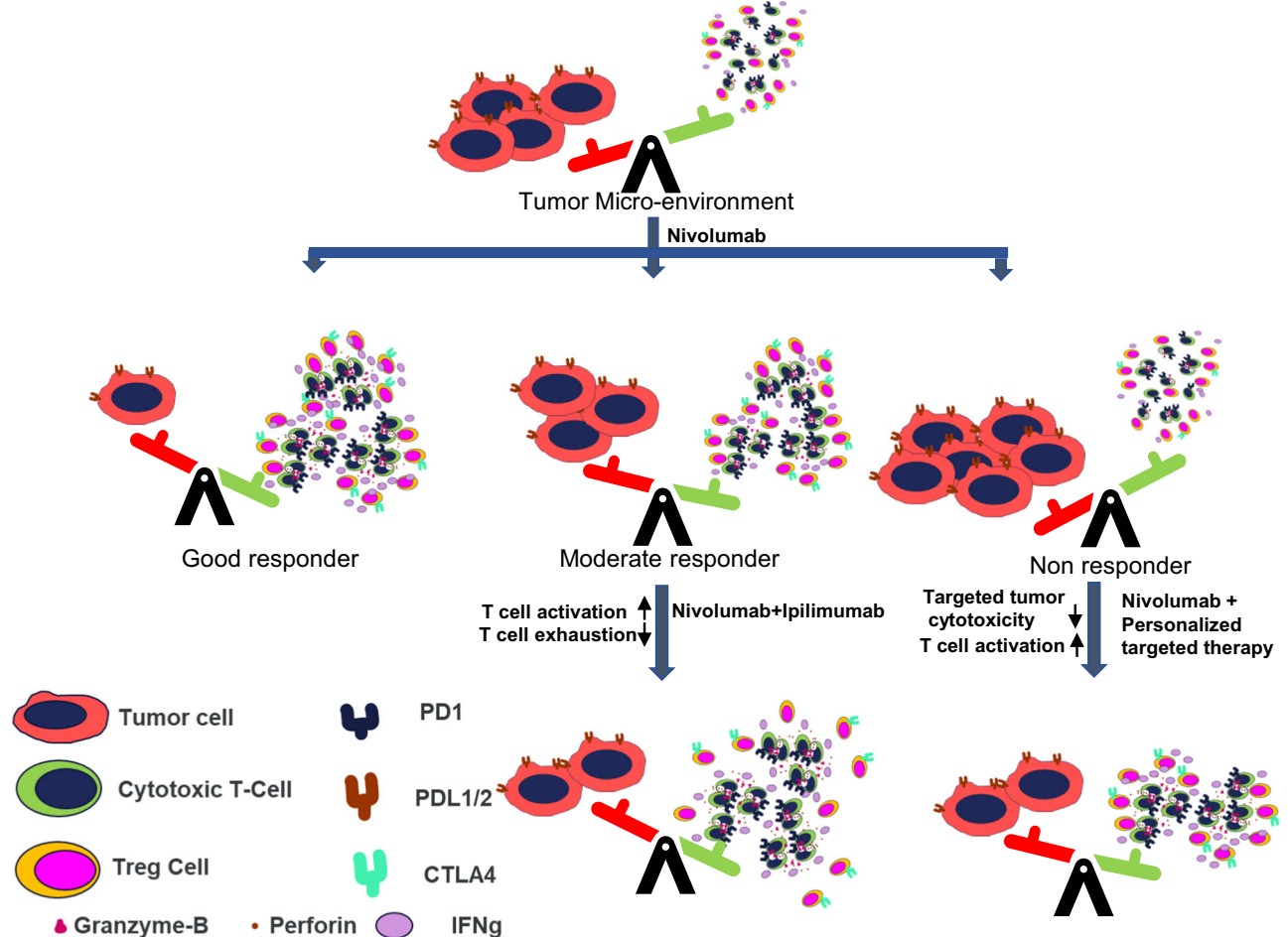

**Fig. 9 | Model elucidating the value of a TruTumor like platform in aiding personalized anti-PD1/PDL1 treatment regimen decision making.** Response to anti-PD1 treatment is determined by the microdynamics between immune cell and tumor response. Immune response against tumor is driven mainly by CTLs and are modulated by immunosuppressive Tregs. Response to treatment is driven by the CTLs overcoming the Tregs and tumor intrinsic factors. Identifying patient specific immune and tumor factors would help choose the appropriate combination therapy options to improve treatment outcomes.

## Cytokine analysis

The cytokine assay was performed for all study arms using the supernatant collected from $T_0$, $T_{24}$, $T_{48}$, and $T_{72}$ using Procarta plex kit (Thermo Fisher Scientific) according to the manufacturer's instruction using 50 μL of supernatant. After the addition of beads, detection antibody, and Streptavidin-Phycoerythin solution, at room temperature, the beads were washed, and Magpix sheath fluid was added, and plates were run on the MAGPIX instrument. We studied 18 cytokines, namely IL-1 beta, IL-2, MMP-9, IP-10 (CXCL10), IL-6, IL-8 (CXCL8), Fractalkine (CX3CL1), GCSF (CSF-3), GMCSF, I-TAC (CXCL11), MCP-1 (CCL2), MIG (CXCL9), M-CSF, IL10, IFNg, TNFa, Perforin and GranzymeB. Of the 18 cytokines IFNg, Perforin and GranzymeB release was assayed for all the 70 Nivolumab treated samples. Analyst software was used for generating pg/ml data. For treatment comparison, the pg/ml data for each time point was normalized using the $T_0$ data for the corresponding cytokine to prevent any variation between the arms for the functional cell types. Fold change analysis was performed in Excel.

## Histopathology

Hemotoxylin and Eosin (H&E) staining was performed on 4 μm thick FFPE sections using a Multi-Stainer (Leica ST5020) using hematoxylin (Merck) and eosin (Eosin-Y, Merck). Leica Biosystems' Aperio AT2 slide scanner was used to scan slides at a 20x magnification, and data were then uploaded to E-Slide management for digital pathology analysis.

Validated positive and negative controls were used in every run. Pathologists scored percentage of tumor content in the tissue fragments as 100 x (tumor area/total fragment area): immune content in the tissue fragment as 100 x (Area covered by immune cells/total fragment area): immune infiltration as 100 x (Area covered by mononuclear immune cells around the tumor nest/total stromal region in the fragment): pyknosis and discohesion scored as a range between 1-5 where, 1 = 1-10%, 2 = 11-25%, 3 = 26-50%, 4 = 51-75% and 5 = 76-100%) and tumor grade[35]. Tumor site/stage, patient age/gender and treatment history were obtained from the sample collection centers.

IHC was carried out on sectioned tumor samples on a BenchMark XT instrument (Roche Diagnostics). Sections were subsequently stained for 12 minutes at 37 °C with antibody against cleaved caspase-3c (Biocare) or CD8 (Roche). 3,3'-diaminobenzidine (DAB) (ultraView Detection Kit; Ventana Medical Systems) was used for signal detection. Hematoxylin-II and Bluing Reagent (Ventana Medical Systems) were used to counterstain the slides. Aperio AT2 slide scanner (Leica Biosystems) was used to scan slides at a 20x magnification, and images were then uploaded to E-Slide management for digital pathology analysis. Certified Pathologists scored the percentage of tumor containing cleaved caspase-3 (100 x (Number of cleaved caspase-3 positive tumor cells/ Total No. of viable tumor cells)) or CTLs stained with CD8 within the tumor or in the stroma. PD-L1 (clone:22C3, Dako) IHC was performed and scored as Combined Positive Score (CPS = 100 x (Number of PD-L1 positive cells-tumor cells, lymphocytes,

macrophages)/ Total number of viable tumor cells). Antibody details are provided in Supplementary Table 3.

## Multiplex IHC analysis

Multiplex IHC was done manually using 4 μm FFPE slices on post treatment samples using Opal dyes (Akoya Biosciences) or TissueGnostics Asia-Pacific using TSA mIHC kits. Markers FoxP3 (1:200, Abcam), CD8 (RTU, Roche or 1:1000, Abcam) CD4 (1:100, Abcam) and panCK (RTU, Dako or 1:1000, Abcam) along with DAPI nuclear stain was used. Data was acquired using Zeiss Axio observer7 or TissueFAXS SPECTRA (TissueGnostics, Austria) and analyzed using QuPath[36] software. All the samples were characterized and quantified using cell detection and pixel classification-based tumor annotation tools. The CTL and Treg counts and their distance to each other and the tumor region was measured using cell classification and 2D spatial analysis tools respectively. Antibody details are provided in Supplementary Table 3.

## RNA extraction and NanoString analysis

RNA was extracted from 8-10 FFPE curls of 4 μm thick using Qiagen RNeasy FFPE mini kit. The sections were deparaffinized using Deparaffinization solution (Qiagen) at 56 °C for 3 min followed by Proteinase K digestion followed by RNA extraction as per the manufacturer's protocol. RNA quantity and quality was assessed using RNA screen tape (Agilent Bioscience) on a 4200 Tape Station (Agilent Bioscience). 50 ng of RNA quantity based on DV200% was run using IO 360 Pan-Cancer panel on the nCounter (NanoString FLEX) platform. The expression data of 750 tumor and immune-related genes and 20 housekeeping genes were assessed for all study samples across treatment arms.

nSolver 4.0 software was used to normalize expression values using a selected set of housekeeping genes and advance analysis (Pathway scores, cell type scores, differential expression analysis) was performed on normalized data. The normalized data was used for gene expression signatures (GES) analysis. The details of the GES are provided in the Supplementary Table 1. The GES was derived by averaging the Log2(m-RNA counts) of each gene.

## HPV detection

DNA was extracted from fresh tissue using ReliaPrep™ gDNA Tissue Miniprep System (Promega), according to the manufacturer's protocol. DNA concentration and purity were evaluated spectrophotometrically. Two different pairs of oligonucleotides primers were used in PCR for amplification of L1 gene of HPV: MY09 (5′-CG TCCMARRGGAWACTGATC-3′)/MY11 (5′-GCMCAGGGWCATAAYAAT GG-3′), GP5+ (5′-TTTGTTACTGTGGTAGATACTAC-3′)/GP6+ (5′-GAAA AATAAACTGTAAATCATATTC-3′) along with β-globin gene (GH20: 5′-GAAGAGCCAAGGACAGGTAC-3′, PC04: 5′-CAACTTCATCCACGTTCA CC-3′) as internal control. DNA isolated from HeLa cell lines was used as positive test control. The PCR protocol[37] used is as follows: 500 ng of extracted DNA was added to 20 μl PCR master mix (TaKaRa Taq™ DNA Polymerase kit) with 0.5 pmol/μl primers for MY09, MY11, GP5 +, GP6 +, and 0.2 pmol/μl primer for PC04, and GH20 and supplemented with 5% DMSO. After Initial denaturation at 94 °C for 5 min annealing was performed at 56 °C for 1 min (MY09/11 & PC04/GH20), and 40 °C for 40 sec (Gp5 +/GP6 +). The PCR for DNA detection was run for 40 cycles and the final extension was done for 5 min at 72 °C.

## t-SNE & statistical analysis

The unsupervised nonlinear dimension reduction method t-distribution–based stochastic nonlinear embedding (t-SNE)[38] was applied to the multimodal data to investigate in two-dimensional space how tumor samples from different HNSCC patients were in relation to each other. Ten parameters: IFNg, IL10, TNFa, Perforin,

and GranzymeB secretion, tumor content, immune content, and immune cell infiltration from H&E and cleaved caspase-3 expression in tumor based on pathologist assessment and tumor fragment viability using PrestoBlue assay for this analysis. Data were first converted to Log2 fold change with respect to the control. The parameters used to generate the tSNE were as follows: perplexity: 10, learning rate: 100 and number of iterations: 1000. The algorithm to generate tSNE was developed in collaboration with Watershed Informatics (Boston, USA). All data analysis and graphical representations were done using GraphPad Prism (Version 9). Wilcoxon matched-pairs signed rank t-test for paired data and Mann-Whitney t-test for unpaired data, was used to generate P-values. Differential gene expression analysis for NanoString data was performed using nSolver 4.0 and Wald test was used to calculate p-value significance. Log-rank (Mantel-Cox) test was used to calculate p-value in Kaplan Meier curve. P-value significance is represented as * ($p < 0.05$) ** ($p < 0.01$), *** ($p < 0.001$), **** ($p < 0.0001$). P-values are mentioned for near significant data (0.05-0.08) and omitted if $p > 0.08$. Heat maps were generated on GraphPad. Hierarchical analysis using Euclidean distance method was performed with Morpheus software (https://software.broadinstitute.org/morpheus).

## Gene signature curation for Nivolumab response prediction

Differential gene expression analysis of SC1 and SC2d samples was done using the NanoString nSolver 4.0. From the result matrix, 55 genes were found to be significantly altered (p-value less than 0.05) (Supplementary Fig. 7B). We performed Principal Component Analysis (PCA) using the expression matrix of these 55 genes. The PC1 and PC2 with percentage of explained variances of 39.11% and 18.52% respectively were able to segregate SC1 and SC2d samples into two clear clusters (Supplementary Fig. 7C left panel). From the knee plot of loading scores, 19 genes were identified as having a loading score of PC1 more than 0.7. (Supplementary Fig. 7C right panel). Next, these genes were compared with Foy et al.[39] baseline HNSCC cohort CLB-IHN clinical data set to obtain a 12 overlapping gene set. Gene expression signature (GES) for these 12 genes was calculated by deriving the average of the Log2 normalized mRNA counts. The methodology followed is schematically described in Supplementary Fig. 7A. In the context of developing an on-treatment gene signature, a comparative analysis of differential gene expression after treatment was performed in nSolver 4.0 software, between the SC1 and SC2d sub-cohorts (Supplementary Fig. 8B). This resulted in a list of 78 significantly altered genes (p-value < 0.05). Of these 78 genes, 31 exhibited a statistically significant difference in expression levels when comparing responders to non-responders within the metastatic melanoma clinical dataset as reported by Chen et al.[20]. GES score for these genes was calculated by computing the average of Log2 Normalized mRNA counts. The methodology followed is schematically described in Supplementary Fig. 8A.

## Reporting summary

Further information on research design is available in the Nature Portfolio Reporting Summary linked to this article.

## Data availability

The NanoString data generated in this study are submitted to GEO database with accession GSE233980, GSE234136, GSE234138. Publicly available Foy et al.[39] clinical data set for CLB-IHN cohort used in this study is available in the GEO database under accession code GSE159067. Chen et al.[20] clinical data set used in this study was obtained from the manuscript 's Supplementary Tables 1-11 (https://doi.org/10.1158/2159-8290.CD-15-1545). All the remaining data are available in the Article, Supplementary Information and Source Data file. Source data are provided as a Source Data file. Source data are provided with this paper.

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

## Acknowledgements

All patients who contributed their valuable samples towards this research work. Liu Xiao Jing and her team from TissueGnostics Asia-Pacific for their technical support in generating some of the multiplex IHC data.

## Author contributions

N.P.B. designed and executed the data analysis strategy and wrote the manuscript. K.J. compiled data and performed critical experiments. N.P.B., K.J., B.D., O.M., R.S.M., V.K., and S.G. designed and executed assays. S.V., C.B., U.K., and G.S.K executed assays. R.M. performed histopathology evaluation. M.N, J., K.R. and A.S. contributed towards data analysis. G.M.S., A.P.S., P.B.V., V.P., J.P.C. and M.B.V., provided HNSCC samples and clinical guidance, M.M. created the collection center network and provided critical comments on the manuscript. G.K. coordinated the collection of samples and clinical data. S.S. conceptualized, supervised the study, and critically reviewed the manuscript.

## Competing interests

The authors from Farcast Biosciences India Pvt. Ltd are current (NPB, KJ, BD, OM, RSM, SV, CB, UK, RM, MN, J., KR, AS, MM, GK, SS) or former (SG, VK, GSK) employees of the company. The remaining authors declare no competing interests.
