## [Peer Review File · Nature Communications]

Tumor histoculture captures the dynamic interactions between tumor and immune components in response to anti-PD1 in head and neck cancerEditorial Note: Parts of this Peer Review File have been redacted as indicated to maintain the confidentiality of unpublished data.

REVIEWER COMMENTS

Reviewer #1 (Remarks to the Author): with expertise in HNSCC, cancer immunotherapy

In the study entitled “Capturing the dynamic interactions between tumor and immune components of the tumor micro-environment is important for determining efficacy of anti-PD1 treatment” the authors describe a new method to culture tumor explants ex vivo while preserving certain aspects of the tumor microenvironment, such as immune cell composition. Furthermore, the authors apply their method to study response to immune checkpoint blockade. The method developed by the authors could potentially be useful to study treatment-specific responses ex vivo, however, this work still has several shortcomings, which should be addressed before publication.

Remarks

In general, although a relatively large number of 89 samples was available in this study many of the conclusions were made based on small subsets of samples. In addition, the selection criteria of the corresponding samples are unclear. A more systematic and comprehensive characterization of the histoculture method developed here should be performed. Furthermore, selection criteria for samples, used to draw conclusions about treatment responses, should be clarified.

1)

In lines 76-77 the authors state that samples with average tumor content less than 10% at baseline were excluded. How many samples were effected? It seems as if all 89 samples were included in the study (19 + 70). It would be good to clarify.

2)

The analysis of intratumoral immune cell preservation is difficult to follow and inconclusive.

Why were only four samples analyzed/shown? How were these samples selected? How were the cell types in Fig S2 determined? What data was used in this analysis (flow cytometry)? Also, it is not obvious from Fig S2B what the cell type fraction of each sample pre- and post-culture. The authors should consider a different representation of the data to clearly demonstrate a reasonable correspondence between cell type proportions pre- and post-treatment (e.g. line graphs). Furthermore, as this is an important aspect of the histoculture developed here, this analysis should be performed comprehensively and not just on four samples.

3)

The authors state that the histoculture platform was optimized with regard to number, size and culture conditions of explants. However, there is very little information regarding the optimization process. For instance, it is not clear why seven explant slices were chosen. Why are mRNA counts an indicator for arm equivalence? Also, what data is shown in Fig S3D? Is this one sample only? It is not clear how the samples were selected that are shown in Fig. S3. For instance, Fig. S3A and S3C only have one patient sample in common. Why is this analysis not systematically performed on all 19 samples that were used for histoculture standardization? Similarly, why are there only eight samples in Fig S3E? How were they selected? These samples do not seem to be representative when compared to Fig 2A. This should be clarified.

It would be helpful to include a more comprehensive analysis of all baseline samples. E.g. a heatmap showing cell type proportions to see if there are clusters of samples with similar TME composition. This would be a good basis to demonstrate that certain sample characteristics can be preserved during culture.

4)

Similarly, the analysis shown in Fig S4 should be expanded. Is this only one sample? Which one and why this one? How was the Teff/Treg ratio determined in Fig 4SB? The authors should perform a more comprehensive characterization of their TruTumor platform.

5)

In Fig 2E, the authors should state how the data for each treatment condition was summarized (e.g. mean expression); also it should be clarified how p values were obtained.

6)

The authors identify distinct sample subsets (SC1, SC2a-d) that can be distinguished based

on treatment response considering T cell reinvigoration (increase IFN γ levels) or tumor cytotoxicity (tumor content decrease, cleaved Caspase-3, GZMB and PRF1 release). SC1 samples displayed largest increases in IFN γ protein and transcript levels, perforin and a concomitant reduction in tumor content. It would be helpful to provide more comprehensive characterization of the baseline samples to potentially explain the differences in treatment response. Are the samples of SC2d depleted of immune cells? Although the authors evaluate differences for a subset of immune cell populations, it would be important to perform a more comprehensive analysis.

7)

The assignment of genes to functional categories in Figure 4E requires further support. For instance, I am not aware of publications relating CCL7 to antigen presentation; IL11RA as marker for cell proliferation. CD3D is also expressed by naïve T cells. The authors should expand the discussion of the cell types and provide references verifying the assigned functions of the corresponding genes.

8)

How was the pathway analysis in Fig 7A performed? Also, what does the color scale indicate?

9)

The authors identify 32 differentially expressed genes the two cohorts. However, in Fig S6 it is shown that the differential gene expression analysis only involves three samples of each subset. How were these samples selected? Many of the 13 selected genes are involved in chemo/cytokine signaling and are immune-related genes. Is it possible that this signature only distinguishes immunologically 'hot' from 'cold' tumors, which are known to respond differently to immunotherapy?

10)

In Figure 8C the authors relate Treg counts to decrease in tumor content and suggest that response to combination therapy is associated with a decrease in Treg to CTL distance. The bar graphs on the right show CD8:Treg distance for the two samples discussed in the main

text and CD8:Tumor distance for the third sample. Was this mislabeled? If not, what is the reason for this choice? The authors should include the same plot. Also, three is a very small sample size. It would be helpful to include additional samples to further support that there is an association between change of CTL:Treg distance and treatment response.

Minor points

- In the introduction the authors refer to granzyme B and perforin as cytolytic cytokines, which is inaccurate, as these proteins are no cytokines.
- The authors should also reconsider the use of the word 'anergy' in the introduction.
- The units in Fig S3B-E are missing.
- Line 98: remove one 'release'
- 55 patients were selected for response analysis. How were they selected?
- The authors refer to IFNg GES. A gene expression signature usually involves multiple genes. It would be more accurate to refer to IFNg expression.
- The color scales in Figs 4E-F don't seem to be accurate. Same in Figs 7A and E.
- The authors should state what they mean by (PR/CR) and (SD/PD)
- The authors should clarify what is meant by PD-L1 CPS.
- Line 407 is partially repeated.
- Line 431: why do the authors refer to CD68 as intracellular marker?

Reviewer #2 (Remarks to the Author): with expertise in HNSCC, cancer immunotherapy

The study entitled "Capturing the dynamic interactions between tumor and immune components of the tumor micro-environment is important for determining efficacy of anti-PD-1 treatment" addresses an important question, which is the absence of surrogate or predictive markers of response to immune checkpoint blockade (ICB) in most cancer types. In this study, the authors focused on head and neck cancer (HNC), and developed a patient derived TruTumor histoculture platform and investigated the response spectrum of patients to anti-PD-1 treatment. Based on the responses in the TruTumor model, they stratified patients based on level of T-cell reinvigoration and tumor cytotoxicity, into 5 sub-cohorts. Perhaps as expected, the responder cohort exhibited high T-cell reinvigoration, high tumor cytotoxicity with T-cells honing into the tumor upon treatment. The worst responder cohort

instead exhibited immune suppression and tumor progression pathways. They then explored the possibility of combining anti-CTLA-4 and anti-PD-1 in the intermediate responders, showing an increased response as judged by improved Cytotoxic T-cell: T-regulatory cell ratio and enhanced tumor cytotoxicity. The baseline and on treatment gene expression signatures derived from this study was able to stratify responders and non-responders in unrelated clinical datasets.

The strength of the study is the development of the TruTumor ex-vivo preclinical platform representing the complexity of the tumor microenvironment (TME) in which immune oncology agents, including ICB, can be tested. The model is elegant and quite relevant. The major weakness is that most of the findings (including the gene expression signatures - GES) are aligned with expected results from the already known effects of anti-PD-1 (and anti-CTLA-4), and hence the authors have not improved our understanding of an effective response to ICB. In addition, they have not validated whether the response in the TruTumor model predicts the ICB response in the patients, nor they consider the clinical observation that combination of anti-CTLA-4 with anti-PD-1 has not improved the response to anti-PD-1 in multiple clinical trials in HNC. In addition, not until the very end the authors describe current biomarkers used for patient stratification, and do not describe prior studies in which an interferon signature was identified as a predictive biomarker in HNC in response to anti-PD-1 ICB. These, and multiple additional issues (see below) make this study descriptive in nature, and more based on an optimized and elegant technical platform than on any specific new discovery.

Additional comments:

Patient selection: Most patients include low grade tumors and stage II/III HNC cases, and hence patients do not have metastatic lesions. It is not clear if this is the population most likely to receive ICB as compared to other standard of care treatments. In general, most responses occurred in the low-grade tumors, which is of interest but likely expectable.

The interferon response described here is likely expectable based on prior studies in which an interferon signature was identified as a predictive biomarker in HNC in response to

pembrolizumab, a different anti-PD-1. Indeed, CXCL10 and other genes are part of the interferon signature often used by Merck to stratify patients, in addition to the PD-L1 CPS >1.

In this regard, not surprisingly, the SC1 sub-cohort (best responders) contained more samples with PD-L1 CPS score of >50, and lower histological grade (78% are grade 1).

While many GES were identified, the interferon signature appears to be the dominant signature. That said, many of the claimed signatures are really represented by a handful (very often a single) gene.

The authors include in the discussion a long laundry list of genes based on GES and individual genes correlating with responses, and they speculate on the potential role of these genes and signatures controlling the TME and ICB response. This is not helpful unless they conduct an experimental validation of key new predictions. In this case, it is difficult to discern what are the new discoveries of the study vs what has already been reported.

Unless mistaken, 12 of the proposed predictive 16 gene set were common to a metastatic melanoma dataset, a very different population than in the current study.

Reviewer #3 (Remarks to the Author): with expertise in HNSCC, cancer immunotherapy

The current manuscript involves very interesting data from a TruTumor HNSCC histoculture platform, treated with nivolumab, with the ultimate aim to find predictive biomarkers for in-human aPD1 response. Overall, in my view, ex-vivo histocultures (not being organoids as they lack immune cells) are the way to move forward in the field and I have read the manuscript with great interest. Although the TruTumor concept and some of the data are quite convincing, I do have some concerns about the selection of patients and biopsies (selection bias?) and also about the number of samples (ranging from 3 to 55) that have been investigated in view of the quite strong conclusions that have been drawn.

1. The word treatment is used throughout the manuscript. Please describe in the manuscript whether you are relating to 'ex-vivo treatment' in your histoculture or data from in-vivo clinically treated patients, both in text and legends.

2. Patient and biopsy selections are not clear. Please add patient numbers and timing of biopsies (in view of clinical treatment) to F1A. (And what is high content imaging?)

- 19 patient biopsies (F1B) were used to establish the platform, however, only a small nr of 4 samples were used in SF2 to show that the immune compartment is maintained during the 72 hour ex vivo assay, and in SF3 only a very small number of 3 samples and 14 samples were used to explore arm equivalence. Please provide all data for all 19 samples to prevent selection bias.

- In Table 1, not all data sum up to 89. What does the category recurrent mean, as also primary tumors are involved? These primary tumors were not treatment-naive? please add previous treatments and TNM stage in Table 1. What kind of immunotherapy was given to these patients? Please specify in Table 1.

- Line 76: How many samples were excluded? How many were used for the ex-vivo TruTumor assay after all?

- the text refers to 'unrelated clinical datasets (more than 1?). Where are these datasets described and when were these used for validation?

- Line 118: A sub-cohort of 55 (patients or biopsies?) was employed. Why and how was this selection made? How many clinical Rs and NRs were there? What is actually meant by 'true Rs or NRs' at this time: clinically or ex-vivo?

At the end, the investigators conclude there are 9 SC1 true responders and 6 SC2d true NRs, which makes a total of 16. Please clarify the numbers 55 and 16.

- My suggestion would be to enlarge the number of HNSCC patients for validation of the signatures found in the TruTumor culture to above 100 patients.

3.

How is arm equivalence defined?

@ R46: 'While maintaining equivalence across arms, F1B'. Could the investigators provide data that this equivalence was reached in (part of) the 19 samples?

How exactly was tumor heterogeneity addressed? Data?

Were multiple biopsies taken per patient?

What was the optimal size of tumor fractions? The optimal nr of replicates per fraction? The plating strategy? Data?

4. Why was the vibratome used?

5. What is immune cell functional 'fidelity'? And what is the rationale to investigate this specifically via aCD3IL2 and LPS as shown in F1C? Were all 19 samples used here? If not, why not?

What compound was used for 'myeloid stimulation'? Could the authors please provide all data?

6. SF2: It seems that mainly macrophages are preserved after the 72 hour ex-vivo culture? Do the authors think that this may have biased their results?

7. In Figure 2, the effect of NIVO on biopsies in the TruTumor platform is shown.

- Are all 19 samples involved? If not, why not?

- To better understand what happens before, during and on-treatment in treated and control arms, could the investigators transform figures 2b-c-d into box plots (with individual datapoints per sample) at baseline and on-treatment, and provide p-values to mark significant changes per group?

8. Figure 3 shows very interesting and convincing data. However, please explain why only 55 patients (biopsies?) were used, while 70 patients were available. Where are the data from the other 15 patients? Why were they excluded?

Was it possible to successfully TruTumor all patients or not, due to contamination, necrosis etc? Could the investigators present the success rate of the TruTumor assay?

9. F3A: There seems to be 1 patient sample that is closer to 5/SC1 than to SC2. Why / how was the cutoff set at 5?

10. For the flow of readability and interpretation, please add in the legends of F3B the words/terms 'tSNE_1' and 'tSNE_2' behind their phenotype. Thank you.

11. F3C: The investigators use IFN γ release as a surrogate marker for T cell reinvigoration. My suggestion would be to keep it to the raw data here and refer to a potential (!) T cell reinvigoration in the text of the manuscript at time of discussing the potential immune cell dynamics behind the signatures in the text paragraph. The same counts for F3E.

12. How many samples were used for Figure 4? And how were these selected?

F4C: To better understand what happens before, during and on-treatment in treated and control arms, could the investigators transform figures 2b-c-d into box plots (with individual datapoints per sample) at baseline and on-treatment, and provide p-values to mark significant changes per group?

13. Line 169-176: For spatial distribution only 1 R and 1 NR were selected. Can the investigators provide data for all SC1 and all SC2d subcohorts here? Is there a gradient from SC1 > SC2 a > SCb > SCc > SCd?

14. Lines 159-187 are discussing biomarkers for response in the Tru Tumor platform. What is the meaning of lines 190-199? To validate these biomarkers on clinical response? please make clear in the title of this paragraph. For me, lines 190-199 partly belong to the paragraph above and partly to the paragraph below...

15. Starting line 201, the ex-vivo signatures found in the TruTumor cohort are evaluated as a predictive biomarker for clinical response, right? Please change the title of this paragraph accordingly.

- Line 218: 10 of 13 Nanostring genes were selected to segregate SC1 and SC2d. In SF6D there are 6 subcohorts. Which is the 6th one? (SC1, SC2a,b,c,d = 5).

- Could the investigators also show data from the other 2 nanostring genes (and potentially discuss why the expression of these genes is not correlated to response or not relevant for our clinical cohort?).

16. Great idea to test NIVO/IPI on the truTumor assay, as we may indeed expect a direct effect of IPI on the TME itself. How many samples were investigated for the NIVO/IPI cohort F*A and B? I would like to suggest not to use the term Treg depletion in this TruTumor setting, as where would they go?? Please keep it to the raw data, Nanostring Treg signature (could change in activity of Tregs?) and spatial distribution.

Could the investigators test IPI only on another 3 samples and investigate C, NIVO, IPI, and NIVO/IPI?

Reviewer #4 (Remarks to the Author): with expertise in HNSCC, histocultures

Key results

The authors of the manuscript entitled Capturing the dynamic interactions between tumor and immune components of 1 the tumor micro-environment is important for determining efficacy of anti-PD1 2 treatment intend to study response to immune checkpoint inhibitors in a preclinical explant model for HNSCC. Patients could be sub-stratified on the basis of T-cell reinvigoration and tumor cytotoxicity into responders and non-responders with graduations of the features evaluated.

Validity

Your evaluation of the validity and robustness of the data interpretation and conclusions. If you feel there are flaws that prohibit the manuscript's publication, please describe them in detail.

The authors based their study on a robust and impressive case number of 70 tumor samples, The methods applied are extensive and appropriately applied.

However, there are some points that need to be addressed to back the translation ability into clinical routine:

1) The samples were cultured for 72 hrs. Authors should substantiate why they chose this cultivation time. As it is accepted that resistance to anti-tumor treatments are rather long-term processes it is unclear if 72 hrs are sufficient to make a statement on treatment response.

How are early and late response defined in this regard (l 383)

2) Have the follow up data and clinical parameters of the donor patients compared with the experimental data regarding response to treatment (anti-checkpoint and other agents)?

3) Why were the histocultures treated with Nivolumab instead of Pembrolizumab although the CPS scores were between 3-100% in all samples?

Significance

Your view on the potential significance of the conclusions for the field and related fields. If you think that other findings in the published literature compromise the manuscript's significance, please provide relevant references.

The findings presented here are relevant insofar as predictive markers for sub- stratifying HNSCC patients to immunotherapy are still not available. In this regards, it would be of interest to see a correlation between the follow up data of the donor patients and the experimental outcome of the explants. Did some of the donors undergo checkpoint inhibition and are there data on the respective response/ post-therapeutic follow up?

Data and methodology

Your assessment of the validity of the approach, the quality of the data, and the quality of presentation. We ask reviewers to assess all data, including those provided as

supplementary information. If any aspect of the data is outside the scope of your expertise, please note this in your report or in the comments to the editor. We may, on a case-by-case basis, ask reviewers to check code provided by the authors (see this Nature editorial for more information).

Reviewers have the right to view the data and code that underlie the work if it would help in the evaluation, even if these have not been provided with the submission (see this Nature editorial). If essential data are not available, please contact the editor to obtain them before submitting the report.

Analytical approach

Your assessment of the strength of the analytical approach, including the validity and comprehensiveness of any statistical tests. If any aspect of the analytical approach is outside the scope of your expertise, please note this in your report or in the comments to the editor.

Suggested improvements

1) Clinical and histopathological data and features of the donor collective and the respective samples should be statistically associated with the experimental data (non-responder/responder).

2) Tumor Site Others (Table 1) should be specified as it is nearly 40% of the whole collective. Where only donor tumors originating from the oral cavity included in the trial?

3) Please explain why the proportion of oral cavity HNSCC is nearly 60%. Is there a focus on establishing a model for HNSCC of this sublocalization and what is the rationale to consider oropharynx, hypopharynx and larynx to a lesser extent.

How can the very little percentage of HPV-associated samples be explained?

5) The histopathologic scoring should be described in more detail (I 456)

6) The authors should add a paragraph placing the applicability, robustness, and validity of their model in the context of other similar 3D ex vivo models proposed in the recent literature.

7) The authors should give an explanation why they consider a 72hr cultivation duration as sufficient to appraise therapy response.

Your suggestions for additional experiments or data that could help strengthen the work and make it suitable for publication in the journal. Suggestions should be limited to the present scope of the manuscript; that is, they should only include what can be reasonably addressed in a revision and exclude what would significantly change the scope of the work. The editor will assess all the suggestions received and provide additional guidance to the authors.

Clarity and context

Your view on the clarity and accessibility of the text, and whether the results have been provided with sufficient context and consideration of previous work. Note that we are not asking for you to comment on language issues such as spelling or grammatical mistakes.

The manuscript is well composed and comprehensively structured. The discussion focusses on the own findings. A critical assessment of the model itself in comparison to other 3D explant models is missing, however.

RESPONSE TO REVIEWER COMMENTS:

Pointwise response provided in blue

Reviewer #1 (Remarks to the Author): with expertise in HNSCC, cancer immunotherapy

In the study entitled “Capturing the dynamic interactions between tumor and immune components of the tumor micro-environment is important for determining efficacy of anti-PD1 treatment” the authors describe a new method to culture tumor explants ex vivo while preserving certain aspects of the tumor microenvironment, such as immune cell composition. Furthermore, the authors apply their method to study response to immune checkpoint blockade. The method developed by the authors could potentially be useful to study treatment-specific responses ex vivo, however, this work still has several shortcomings, which should be addressed before publication.

Remarks

In general, although a relatively large number of 89 samples was available in this study many of the conclusions were made based on small subsets of samples. In addition, the selection criteria of the corresponding samples are unclear. A more systematic and comprehensive characterization of the histoculture method developed here should be performed. Furthermore, selection criteria for samples, used to draw conclusions about treatment responses, should be clarified.

Samples were selected based on the 10% baseline tumor content qualification criteria. Once included, samples were run on multiple assays based on the availability of sufficient tissue. For example, duplicate arms are required to run histopathology and flow cytometry experiments, multiple curls of FFPE tissue are required for isolating RNA to run the Nanostring assay leaving with limited tissue sections to run the basic histopathology and mIHC assays. This was the main reason for not running all experiments on all samples. In addition, with the limited funding and reagents available to complete this study, we had to restrict the number of samples that could be run on the cytokine assay For generating the sub-cohorts using the PCA/tSNE approach we included the first 55 samples for which we had generated data for all the 10 parameters used for cohort segregation.

1)

In lines 76-77 the authors state that samples with average tumor content less than 10% at baseline were excluded. How many samples were effected? It seems as if all 89 samples were included in the study (19 + 70). It would be good to clarify.

All samples included in this study had more than 10% tumor content at baseline. Based on all samples that we recruit from various centers; we see a rejection rate of 19%. (Line 82-83)

2) The analysis of intratumoral immune cell preservation is difficult to follow and inconclusive. Why were only four samples analyzed/shown? How were these

samples selected? How were the cell types in Fig S2 determined? What data was used in this analysis (flow cytometry)? Also, it is not obvious from Fig S2B what the cell type fraction of each sample pre- and post-culture. The authors should consider a different representation of the data to clearly demonstrate a reasonable correspondence between cell type proportions pre- and post-treatment (e.g. line graphs). Furthermore, as this is an important aspect of the histoculture developed here, this analysis should be performed comprehensively and not just on four samples.

The selection of four samples for immune cell preservation was purely based on available sample quantity. Based on the feedback, we have included additional samples (n=9) to study immune cell preservation. The revised data has been included in the manuscript. Line graph representations have been included to show preservation of various cell sub-types (Suppl Fig 2C).

3) The authors state that the histoculture platform was optimized with regard to number, size and culture conditions of explants. However, there is very little information regarding the optimization process. For instance, it is not clear why seven explant slices were chosen. Why are mRNA counts an indicator for arm equivalence? Also, what data is shown in Fig S3D? Is this one sample only? It is not clear how the samples were selected that are shown in Fig. S3. For instance, Fig. S3A and S3C only have one patient sample in common. Why is this analysis not systematically performed on all 19 samples that were used for histoculture standardization? Similarly, why are there only eight samples in Fig S3E? How were they selected? These samples do not seem to be representative when compared to Fig 2A. This should be clarified.

Details behind the logic of selection of explant replicates per arm has been included in the manuscript (lines 107-112, Suppl. Fig 3). We compared the level of expression of genes (normalized mRNA counts) between arms to establish arm equivalence with the assumption that equivalent regions should have equivalent gene expression patterns. Suppl. Fig. 4C with the mRNA expression data has been appropriately labelled.

As mentioned earlier, not all assays could be performed on all samples due to sample size limitations. We have included additional samples (n=5) to strengthen arm equivalence data in the manuscript (lines 117, Suppl. Fig 4D).

The samples are selected based on availability of explants and not biased. Wherever possible we have generated assay level data and added it to the manuscript.

The 8 samples are part of 19 samples used for platform establishment and not part of the 70 samples represented in Fig 2A.

It would be helpful to include a more comprehensive analysis of all baseline samples. E.g. a heatmap showing cell type proportions to see if there are clusters of samples with similar TME composition. This would be a good basis to demonstrate that certain sample characteristics can be preserved during culture.

Cell type data represented as heatmap has been included in the manuscript (Lines 96-99, Suppl. Fig. 2D4). There was no obvious correlation between proportions of different immune cell types and preservation levels.

Similarly, the analysis shown in Fig S4 should be expanded. Is this only one sample? Which one and why this one? How was the Teff/Treg ratio determined in Fig 4SB? The authors should perform a more comprehensive characterization of their TruTumor platform.

The analysis shown in S4 was from a representative sample. Analysis from 3 samples is provided in Figure 1C. Teff/Treg ratio was calculated from mRNA expression data using gene expression scores for these two cell types.

5) In Fig 2E, the authors should state how the data for each treatment condition was summarized (e.g. mean expression); also it should be clarified how p values were obtained.

The data was generated using 34 samples for which NanoString was performed for Nivolumab treatment. Using nSolver advanced analysis software differential gene expression was performed with the control and nivolumab treated samples and represented as a volcano plot. The Log₂ fold change and p-values were obtained from this analysis. This is briefly mentioned in Fig 2E legend and has been added to the methodology in the manuscript (line 606-607).

6) The authors identify distinct sample subsets (SC1, SC2a-d) that can be distinguished based on treatment response considering T cell reinvigoration (increase IFN γ levels) or tumor cytotoxicity (tumor content decrease, cleaved Caspase-3, GZMB and PRF1 release). SC1 samples displayed largest increases in IFN γ protein and transcript levels, perforin and a concomitant reduction in tumor content. It would be helpful to provide more comprehensive characterization of the baseline samples to potentially explain the differences in treatment response. Are the samples of SC2d depleted of immune cells? Although the authors evaluate differences for a subset of immune cell populations, it would be important to perform a more comprehensive analysis.

Baseline characterization of CD8+CTL content, HPV status, stage and grade is shown in Fig. 6. for these two cohorts. It has been mentioned in line 224 and 225 that the CTLs do not have a significant difference between the two sub-cohorts though SC1 show a higher trend. Baseline gene expression and pathway analysis for all samples from both cohorts has been done and included in Fig. 8 A. SC2d exhibited high gene expression scores for pro-tumor pathways and lower scores for immune activity related pathways. SC2d was not depleted of immune cells. A comprehensive analysis of multiple immune cell gene signatures across samples at baseline was performed and data is shown in supplementary fig. 2D. Response based segregation was not evident. Immune high and low tumors were present in both SC1 and SC2d sub-cohorts.

7) The assignment of genes to functional categories in Figure 4E requires further

support. For instance, I am not aware of publications relating CCL7 to antigen presentation; IL11RA as marker for cell proliferation. CD3D is also expressed by naïve T cells. The authors should expand the discussion of the cell types and provide references verifying the assigned functions of the corresponding genes.

CCL7 has been shown to be expressed on Dendritic Cells which is why it was assigned to antigen presentation (Zhang M et al. Nat Commun. 2020 Nov 30;11(1):6119.) IL11RA is implicated in tumor progression and hence assigned to cell proliferation pathway (Ernst M, Putoczki TL Clin Cancer Res. 2014 Nov 15;20(22):5579-88.). CD3D was considered as a general T-cell marker and was considered to be involved with T cell activity. We do understand that all these genes have pleiotropic functions, but we have assigned pathways that best fitted with readouts from orthologous assays as well. In the revised version of the manuscript, we have focused more on pathways rather than individual gene function to characterize the role of immune and tumor related activities in determining response to Nivolumab treatment.

8) How was the pathway analysis in Fig 7A performed? Also, what does the color scale indicate?

Pathway analysis was performed as described in the methods section (lines 606),. The figure has been replaced with Pathway scores that are generated by nSolver Advanced analysis instead of gene-based analysis as many genes involved in Immune system has multiple functions.

9) The authors identify 32 differentially expressed genes the two cohorts. However, in Fig S6 it is shown that the differential gene expression analysis only involves three samples of each subset. How were these samples selected? Many of the 13 selected genes are involved in chemo/cytokine signaling and are immune-related genes. Is it possible that this signature only distinguishes immunologically 'hot' from 'cold' tumors, which are known to respond differently to immunotherapy?

Based on the reviewer's feedback gene expression from all samples from both cohorts have been included in the revised manuscript (lines 239-282 , Fig 7, Supp. Fig. 7). Re-analyzed data has been included in the revised manuscript. Additionally, we were able to find a HNSCC anti-PD1 treated clinical data set for generating baseline signature. Based on the re-analyzed data we have identified a gene signature comprising 12 genes. The revised data analysis has been updated in the manuscript (Fig. 7). As mentioned earlier, response signature was not correlated to presence of a "hot" tumor as determined by presence of immune cells but rather with immune cell activity, tumor related signaling pathways, antigen presentation and tumor cytotoxicity (Fig. 7A). However, while generating the gene signature (baseline and on-treatment) we were limited by the gene set used for the clinical dataset that mostly included immune cell related genes. Thus, the possible of role of additional genes identified by the TruTumor platform, beyond those associated with immune cell activity, cannot be ruled out at this point.

10) In Figure 8C the authors relate Treg counts to decrease in tumor content and suggest that response to combination therapy is associated with a decrease in Treg to CTL distance. The bar graphs on the right show CD8:Treg distance for the two samples discussed in the main text and CD8:Tumor distance for the third sample. Was this mislabeled? If not, what is the reason for this choice? The authors should include the same plot. Also, three is a very small sample size. It would be helpful to include additional samples to further support that there is an association between change of CTL:Treg distance and treatment response.

Yes, there was a mis-labeling in the figure. The data has been re-analyzed after including more samples (n=9) from SC2a with varying degree of tumor cytotoxicity. .

Minor points

- In the introduction the authors refer to granzyme B and perforin as cytolytic cytokines, which is inaccurate, as these proteins are no cytokines.

Point has been duly noted and changed to cytolytic proteins in the manuscript

- The authors should also reconsider the use of the word 'anergy' in the introduction.

Anergy has been replaced with exhaustion.

- The units in Fig S3B-E are missing.

The units for fig S3B-E have been added in the revised manuscript,

- Line 98: remove one 'release'

This is corrected.

- 55 patients were selected for response analysis. How were they selected?

- The authors refer to IFNg GES. A gene expression signature usually involves multiple genes. It would be more accurate to refer to IFNg expression.

The first 55 samples were selected for which all 10 parameters were generated for performing the tSNE based analysis. The IFN gamma GES used in this manuscript includes multiple genes. Kindly refer to the details of the gene list provided in the supplementary table (SF1).

- The color scales in Figs 4E-F don't seem to be accurate. Same in Figs 7A and E.

The figures have been replaced with new ones.

- The authors should state what they mean by (PR/CR) and (SD/PD)

We have used the response annotations for complete response (CR, partial response PR, Stable disease (SD) and progressive disease (PD) as mentioned in the corresponding publication from where the data was used. This has also been stated in the manuscript line 261-262. We have not further investigated the criteria for the response category assignment.

- The authors should clarify what is meant by PD-L1 CPS.

PD-L1 CPS (Combined Positive Score) score= $\frac{\text{(No. of PD-L1 positive cells (tumor cells, lymphocytes, macrophages) X 100)}{\text{Total No. of viable tumor cells}}$

This is updated in the methods section of the manuscript (lines 580)

- Line 407 is partially repeated.

The drug concentrations have been mentioned twice to differentiate between mono therapy and combination doses.

- Line 431: why do the authors refer to CD68 as intracellular marker?

To the best of our knowledge, CD68 is an intra-cellular marker. Kindly refer to the reference attached. Br J Haematol . 1995 Aug;90(4):774-82. doi: 10.1111/j.1365-2141.1995.tb05195.x

Reviewer #2 (Remarks to the Author): with expertise in HNSCC, cancer immunotherapy

The study entitled “Capturing the dynamic interactions 1 between tumor and immune components of the tumor micro-environment is important for determining efficacy of anti-PD-1 treatment” addresses an important question, which is the absence of surrogate or predictive markers of response to immune checkpoint blockade (ICB) in most cancer types. In this study, the authors focused on head and neck cancer (HNC), and developed a patient derived TruTumor histoculture platform and investigated the response spectrum of patients to anti-PD-1 treatment. Based on the responses in the TruTumor model, they stratified patients based on level of T-cell reinvigoration and tumor cytotoxicity, into 5 sub-cohorts. Perhaps as expected, the responder cohort exhibited high T-cell reinvigoration, high tumor cytotoxicity with T-cells honing into the tumor upon treatment. The worst responder cohort instead exhibited immune suppression and tumor progression pathways. They then explored the possibility of combining anti-CTLA-4 and anti-PD-1 in the intermediate

responders, showing an increased response as judged by improved Cytotoxic T-cell: T-regulatory cell ratio and enhanced tumor cytotoxicity. The baseline and on treatment gene expression signatures derived from this study was able to stratify responders and non-responders in unrelated clinical datasets.

The strength of the study is the development of the TruTumor ex-vivo preclinical platform representing the complexity of the tumor microenvironment (TME) in which immune oncology agents, including ICB, can be tested. The model is elegant and quite relevant. The major weakness is that most of the findings (including the gene expression signatures - GES) are aligned with expected results from the already known effects of anti-PD-1 (and anti-CTLA-4), and hence the authors have not improved our understanding of an effective response to ICB.

The primary aim of this manuscript was to show how close the response of this ex vivo platform was to clinical data. Hence, we were encouraged by the reproduction of the clinical response on the TruTumor platform.

In addition, they have not validated whether the response in the TruTumor model predicts the ICB response in the patients, nor they consider the clinical observation that combination of anti-CTLA-4 with anti-PD-1 has not improved the response to anti-PD-1 in multiple clinical trials in HNC.

In the absence of clinical validation of the platform, we have tried to correlate response of the TruTumor platform to published clinical response markers. We do appreciate the need to generate clinical validation from matched patients to compare their response to what is observed on this platform. We are currently enrolling for an observational trial wherein patients are being treated with Nivolumab in the clinic and in parallel their samples are being evaluated for response on the TruTumor platform. Early results are encouraging wherein a TruTumor non-responding patient passed away due to progressive disease. Another patient who is predicted to respond as per TruTumor has received six doses and is showing signs of response.

We are aware of the poor response rates in HNSCC for anti-PD1/anti-CTLA4 combination treatment. The purpose of including anti-CTLA4 combination in the current study was to evaluate any additional benefit in patient samples that did not respond well to Nivolumab monotherapy. Response was not uniform for the small set of samples tested.

In addition, not until the very end the authors describe current biomarkers used for patient stratification, and do not describe prior studies in which an interferon signature was identified as a predictive biomarker in HNC in response to anti-PD-1 ICB. These, and multiple additional issues (see below) make this study descriptive in nature, and more based on an optimized and elegant technical platform than on any specific new discovery.

Additional comments:

Patient selection: Most patients include low grade tumors and stage II/III HNC cases, and hence patients do not have metastatic lesions. It is not clear if this is the population most likely to receive ICB as compared to other standard of care treatments.

We appreciate the valid point raised by the reviewer. In the current observational clinical trial, we are testing patients with recurrent disease using tissues from HNC sites. Given that these patients did have aggressive disease that does not respond to conventional SOC treatments, immunotherapy offers a better alternative option making this platform relevant for screening these patients for response.

In general, most responses occurred in the low-grade tumors, which is of interest but likely expectable.

Recurrent tumors do have lower grade and stage and are relevant samples to be evaluated for response on this platform.

The interferon response described here is likely expectable based on prior studies in which an interferon signature was identified as a predictive biomarker in HNC in response to pembrolizumab, a different anti-PD-1. Indeed, CXCL10 and other genes are part of the interferon signature often used by Merck to stratify patients, in addition to the PD-L1 CPS >1. In this regard, not surprisingly, the SC1 sub-cohort (best responders) contained more samples with PD-L1 CPS score of >50, and lower histological grade (78% are grade 1).

While many GES were identified, the interferon signature appears to be the dominant signature. That said, many of the claimed signatures are really represented by a handful (very often a single) gene.

While interferon gamma pathway was picked up as a dominant pathway in our analysis, it did not always correlate with a tumor cytotoxic response. We identify antigen presentation and apoptosis pathways directing response while pro-tumorigenic signaling pathways adversely affecting response (Fig.7A).

The authors include in the discussion a long laundry list of genes based on GES and individual genes correlating with responses, and they speculate on the potential role of these genes and signatures controlling the TME and ICB response.

The discussion has been modified to focus more on significantly modulated pathways rather than focus on individual genes.

This is not helpful unless they conduct an experimental validation of key new predictions. In this case, it is difficult to discern what are the new discoveries of the study vs what has already been reported.

Unless mistaken, 12 of the proposed predictive 16 gene set were common to a metastatic melanoma dataset, a very different population than in the current study.

The point is well taken. In this revised manuscript, we have repeated the analysis using a clinical data set for HNSCC patients treated with anti-PD1 treatment. We could not however, access on-treatment HNSCC dataset, hence have used a melanoma dataset. It is however important to note that even with the difference in indications used for baseline and on-treatment signature development, the significantly modulated pathways represented by these two gene sets are very similar as would be expected from an indication agnostic response from immune check point inhibitors.

Reviewer #3 (Remarks to the Author): with expertise in HNSCC, cancer immunotherapy

The current manuscript involves very interesting data from a TruTumor HNSCC histoculture platform, treated with nivolumab, with the ultimate aim to find predictive biomarkers for in-human aPD1 response. Overall, in my view, ex-vivo histocultures (not being organoids as they lack immune cells) are the way to move forward in the field and I have read the manuscript with great interest. Although the TruTumor concept and some of the data are quite convincing, I do have some concerns about the selection of patients and biopsies (selection bias?) and also about the number of samples (ranging from 3 to 55) that have been investigated in view of the quite strong conclusions that have been drawn.

1. The word treatment is used throughout the manuscript. Please describe in the manuscript whether you are relating to 'ex-vivo treatment' in your histoculture or data from in-vivo clinically treated patients, both in text and legends.

All data presented in this manuscript was generated using an ex vivo platform. This is now stated in the introduction (line 62)

2. Patient and biopsy selections are not clear. Please add patient numbers and timing of biopsies (in view of clinical treatment) to F1A. (And what is high content imaging?)

All samples used in this study were surgical excess tissues. Multiplex IHC was referred to as high content imaging. This has been updated in the revised manuscript in Figure 1A.

- 19 patient biopsies (F1B) were used to establish the platform, however, only a small nr of 4 samples were used in SF2 to show that the immune compartment is

maintained during the 72 hour ex vivo assay, and in SF3 only a very small number of 3 samples and 14 samples were used to explore arm equivalence. Please provide all data for all 19 samples to prevent selection bias.

Only a subset of the 19 samples were of a reasonable size dimension to conduct multiple experiments to establish immune cell preservation. From these 19 samples we have included additional assay data for 3 samples and new additional samples (n=5) have now been added and included in the manuscript.

- In Table 1, not all data sum up to 89. What does the category recurrent mean, as also primary tumors are involved? These primary tumors were not treatment-naive? please add previous treatments and TNM stage in Table 1. What kind of immunotherapy was given to these patients? Please specify in Table.

There was an error in the summation. Thank you for pointing out. The table has been corrected and updated with additional sample information used for extending the preservation experiment data set.

- Line 76: How many samples were excluded? How many were used for the ex-vivo TruTumor assay after all?

A total of 98 samples have been used in this study (revised manuscript) that include 28 for platform establishment and 70 used for drug treatment. Of the 70, the first 55 samples selected without any bias were run on multiple assays from which 10 parameters were evaluated for categorizing the samples into sub-cohorts. This information has been updated in Fig 1B,

- the text refers to 'unrelated clinical datasets (more than 1?). Where are these datasets described and when were these used for validation?

The publications from which these clinical datasets were obtained is referenced in analysis workflow in Supplementary figures 7 and 8 and in methodology line 650 and 670.

- Line 118: A sub-cohort of 55 (patients or biopsies?) was employed. Why and how was this selection made? How many clinical Rs and NRs were there? What is actually meant by 'true Rs or NRs' at this time: clinically or ex-vivo? At the end, the investigators conclude there are 9 SC1 true responders and 6 SC2d true NRs, which makes a total of 16. Please clarify the numbers 55 and 16.

As mentioned, the first 55 patients were selected for running the PCA analysis using data generated from multiple assay readouts. These samples were of optimal size to run these multiple assays. True R's are defined as samples that exhibited bit T cell response (indicated by increased interferon gamma secretion) as well as increase in

tumor cytotoxicity, Non responders (NR) exhibited neither of these two phenotypes. These responses are from the ex-vivo platform and not from patients in the clinic.

Of the 55 samples that were run on PCA analysis to create 5 sub-cohorts. SC1 and SC2d are responders and non-responders from the 55-sample cohort.

- My suggestion would be to enlarge the number of HNSCC patients for validation of the signatures found in the TruTumor culture to above 100 patients.

We have initiated an observation trial that will extend for about an year for patient treatment and response follow-up.

3. How is arm equivalence defined?

@ R46: 'While maintaining equivalence accross arms, F1B'. Could the investigators provide data that this equivalence was reached in (part of) the 19 samples?

How exactly was tumor heterogeneity addressed? Data?

Were multiple biopsies taken per patient?

What was the optimal size of tumor fractions? The optimal nr of replicates per fraction? The plating strategy? Data?

Arm equivalence is defined presence of the level of variability across multiple parameters which when measured across parameters was not statistically significant across untreated or arms.

Multiple assay readouts e.g. Tumor content, Immune content, Ki67 staining in tumor, gene expression level, immune cell sub types and Interferon gamma release response to Nivolumab treatment were used to assess arm equivalence. There was not significant difference between these read outs amongst two replicate arms.

Heterogeneity was addressed by adopting a plating strategy that ensured representation from every cross section of the tumor sample in each arm (lines 106-107).

We received only one piece of tissue per patient. The explant plating strategy employed was to ensure equivalent representation of heterogeneity in explants generated from this single tissue piece.

Each tumor explant is 2-3 mm in length and width and less than 500 microns in thickness. Seven replicates were used per arm and distributed such that each arm had a representative explant from each cross section of the tissue.

4. Why was the vibratome used?

Vibratome is used for generating sub-500 micron sections.

5. What is immune cell functional 'fidelity'? And what is the rationale to investigate

this specifically via aCD3IL2 and LPS as shown in F1C? Were all 19 samples used here? If not, why not?

What compound was used for 'myeloid stimulation'? Could the authors please provide all data?

Functional fidelity was defined by the respective immune cells secreting the expected cytokine upon treatment with a specific stimulant. Anti-CD3 + IL2 was used to stimulate T cells while LPS was used to stimulate the myeloid cells, namely macrophages. Data is presented for three samples. The self-funded study was performed with limited reagents from limited funds. Cytokine assay reagents were optimally used to assess response to Nivolumab treatment after testing for immune cell activity in the TME with three independent samples.

As mentioned above, LPS was used for myeloid stimulation.

6. SF2: It seems that mainly macrophages are preserved after the 72 hour ex-vivo culture? Do the authors think that this may have biased their results?

We do see an overall increase in macrophage population upon culture but the M1 and M2 ratios are maintained. The effect of the drug is evaluated upon comparison with the control arm. The strong interferon gamma response observed on Nivolumab treatment is indicative of presence of active T cells as well. Moreover, treatment specific increase in other immune cells namely NK and T effector memory cells are indicative of limited interference from macrophages.

7. In Figure 2, the effect of NIVO on biopsies in the TruTumor platform is shown.

- Are all 19 samples involved? If not, why not?

- To better understand what happens before, during and on-treatment in treated and control arms, could the investigators transform figures 2b-c-d into box plots (with individual datapoints per sample) at baseline and on-treatment, and provide p-values to mark significant changes per group?

8 samples were used. These samples were selected based on sufficient size required to perform both cytometry and histopathology assays.

Data has now been represented as violin plots with p-values where significant along with individual data points.

8. Figure 3 shows very interesting and convincing data. However, please explain why only 55 patients (biopsies?) were used, while 70 patients were available. Where are the data from the other 15 patients? Why were they excluded?

Was it possible to successfully TruTumor all patients or not, due to contamination, necrosis etc? Could the investigators present the success rate of the TruTumor assay?

The samples were selected without any bias. The first 55 samples after establishing the platform were used to generate 10 different parameters used to sub-stratify samples into various sub-cohorts. A 19% attrition was observed for samples received from centers due to low tumor content. These samples were excluded from the

study. Once recruited all samples qualified on the TruTumor Histoculture.

9. F3A: There seems to be 1 patient sample that is closer to 5/SC1 than to SC2. Why / how was the cutoff set at 5?

The cut-off was arbitrarily set to separate the two clusters. The patient sample lying close to the cut-off belongs to SC2a which is a moderate response sub-group. The true non responders SC2d, however, were well separated from responder SC1.

10. For the flow of readability and interpretation, please add in the legends of F3B the words/terms 'tSNE_1' and 'tSNE_2' behind their phenotype. Thank you.

We are unable to understand this question clearly. The tSNE analysis was performed for all 10 parameters (Fig. 3A). We have integrated the two tSNE plots in the revised manuscript showing all the SC2 sub-stratification. In addition, we have included a table detailing all the phenotype characteristics and showing the number of samples in each sub-cohort.

11. F3C: The investigators use IFN γ release as a surrogate marker for T cell reinvigoration. My suggestion would be to keep it to the raw data here and refer to a potential (!) T cell reinvigoration in the text of the manuscript at time of discussing the potential immune cell dynamics behind the signatures in the text paragraph. The same counts for F3E.

Appropriate change has been made to the figure.

12. How many samples were used for Figure 4? And how were these selected?

F4C: To better understand what happens before, during and on-treatment in treated and control arms, could the investigators transform figures 2b-c-d into box plots (with individual datapoints per sample) at baseline and on-treatment, and provide p-values to mark significant changes per group?

9 out of the total of 15 samples from cohorts SC1 and SC2d have been included in the mIHC experiment based on the availability of tissue sections. Graphs have now been changed to violin plots with p-values.

13. Line 169-176: For spatial distribution only 1 R and 1 NR were selected. Can the investigators provide data for all SC1 and all SC2d subcohorts here? Is there a gradient from SC1 > SC2 a > SCb > SCc > SCd?

The aim of the manuscript is to identify markers that differentiate true responders from true non-responders, hence most data is presented for these two sub-cohorts. Moreover, the spatial data is absolute in nature and a response gradient across multiple sub-cohorts is highly unlikely. We have however shown response gradient

for multiple other parameters like interferon gamma secretion, baseline and on treatment GES.

14. Lines 159-187 are discussing biomarkers for response in the Tru Tumor platform. What is the meaning of lines 190-199? To validate these biomarkers on clinical response? please make clear in the title of this paragraph. For me, lines 190-199 partly belong to the paragraph above and partly to the paragraph below...
The two sections describe on-treatment resistance markers and baseline predictive markers, hence separated into two sub-sections.

15. Starting line 201, the ex-vivo signatures found in the TruTumor cohort are evaluated as a predictive biomarker for clinical response, right? Please change the title of this paragraph accordingly.

- Line 218: 10 of 13 Nanostring genes were selected to segregate SC1 and SC2d. In SF6D there are 6 subcohorts. Which is the 6th one? (SC1, SC2a,b,c,d = 5).
- Could the investigators also show data from the other 2 nanostring genes (and potentially discuss why the expression of these genes is not correlated to response or not relevant for our clinical cohort?).

In the absence of a matched patient clinical response data in the current study, the gene signatures are extrapolated to be predictive in the clinic. Moreover, the signatures are limited by the number of genes that overlap between the TruTumor and the un-related clinical dataset. The two genes that were omitted were absent in the clinical dataset. For these reasons, we would not like to claim this signature to be predictive for clinical response. These two genes are not relevant now since the signature is altered after the re-analysis using additional nanostring data from all SC1 and SC2d samples.

These were six samples belonging to two cohort. We understand that the "cohort" label was misleading. The figure has been changed now, to include all samples from both sub-cohorts.

16. Great idea to test NIVO/IPI on the truTumor assay, as we may indeed expect a direct effect of IPI on the TME itself. How many samples were investigated for the NIVO/IPI cohort F*A and B? I would like to suggest not to use the term Treg depletion in this TruTumor setting, as where would they go?? Please keep it to the raw data, Nanostring Treg signature (could change in activity of Tregs?) and spatial distribution.

Could the investigators test IPI only on another 3 samples and investigate C, NIVO, IPI, and NIVO/IPI?

A total of 14 samples from SC2a sub-cohort were tested for the combination treatment. Anti-CTLA4 antibody is known to deplete Treg depletion by Antibody-dependent cellular cytotoxicity and phagocytosis (ADCC, ADCP). It is in this context that we included this term as TruTumor has all the immune cells that mediate

ADCC/P. We have shown enhanced Teff/Treg ratio using NanoString derived gene expression signatures upon combination treatment. An increase in Teff (CTLs) and a decrease in (Teff) on treatment with the combination contributed to the significant increase in Teff/Treg ratio.

We analyzed three samples for comparing response to Nivo vs Nivo+Ipili vs Ipili alone, as suggested and represented the data below. Data was generated data using flow cytometry, cytokine release and histopathological evaluation of tumor content and cleaved caspase expression within tumor. There was no significant change in Treg proportions between Ipili mono therapy and in combination with Nivo. Ipili monotherapy did not cause an increase in Interferon gamma and Granzyme B secretion. No significant difference in tumor cytotoxicity was observed with any of the three treatments in these three samples. (However, this data is not included in the manuscript)

Ipili data (n=3)

No Significant difference observed between arms for all parameters

Reviewer #4 (Remarks to the Author): with expertise in HNSCC, histocultures

Key results

The authors of the manuscript entitled Capturing the dynamic interactions between tumor and immune components of 1 the tumor micro-environment is important for determining efficacy of anti-PD1 2 treatment intend to study response to immune checkpoint inhibitors in a preclinical explant model for HNSCC. Patients could be sub-stratified on the basis of T-cell reinvigoration and tumor cytotoxicity into responders and non-responders with graduations of the features evaluated.

Validity

Your evaluation of the validity and robustness of the data interpretation and conclusions. If you feel there are flaws that prohibit the manuscript's publication, please describe them in detail.

The authors based their study on a robust and impressive case number of 70 tumor samples, The methods applied are extensive and appropriately applied.

However, there are some points that need to be addressed to back the translation ability into clinical routine:

1) The samples were cultured for 72 hrs. Authors should substantiate why they chose this cultivation time. As it is accepted that resistance to anti-tumor treatments are rather long-term processes it is unclear if 72 hrs are sufficient to make a statement on treatment response.

How are early and late response defined in this regard (l 383)

2) Have the follow up data and clinical parameters of the donor patients compared with the experimental data regarding response to treatment (anti-checkpoint and other agents)?

Beyond 72 h of culture tissue discohesion and pyknosis is seen. Therefore, we fixed 72 h of culture where we observed T cell activation, cytokine release and tumor killing. While acquired resistance might not be apparent within the 3- day culture period, the platform should be able to evaluate non-response phenotype due to an immune-suppressive tumor micro-environment existing in absence of treatment.

Early and late responses were defined based on whether on treatment, just a T cell activity (e.g. Interferon gamma release) was observed versus tumor killing along with interferon gamma release respectively.

None of the patients, whose samples were included in this study have undergone treatment with anti-PD1 treatment.

3) Why were the histocultures treated with Nivolumab instead of Pembrolizumab although the CPS scores were between 3-100% in all samples?

Nivolumab was selected based on availability of cost of the drug. Given the high cost of Pembrolizumab, it is not affordable to patients in low- and middle-income countries. In a study by Tata Memorial Hospital, Mumbai, India, good response was demonstrated on treating HNSCC patients with a low dose Nivolumab and metronomic chemotherapy (Patil, V. M. et al. J Clin Oncol 41, 222–232 (2023)). We are currently running an observational trial using this treatment regimen to test the clinical validity of the TruTumor platform and develop this platform for patient selection.

The findings presented here are relevant insofar as predictive markers for sub-stratifying HNSCC patients to immunotherapy are still not available. In this regards, it would be of interest to see a correlation between the follow up data of the donor patients and the experimental outcome of the explants. Did some of the donors

undergo checkpoint inhibition and are there data on the respective response/ post-therapeutic follow up?

As mentioned earlier, none of the patients included in this study underwent ICI treatment.

1) Clinical and histopathological data and features of the donor collective and the respective samples should be statistically associated with the experimental data (non-responder/ responder).

Grade and stage for samples from both sub-cohorts (SC1 and SC2d) have been shown in figure 7.

2) Tumor Site Others (Table 1) should be specified as it is nearly 40% of the whole collective. Where only donor tumors originating from the oral cavity included in the trial?

3) Please explain why the proportion of oral cavity HNSCC is nearly 60%. Is there a focus on establishing a model for HNSCC of this sublocalization and what is the rationale to consider oropharynx, hypopharynx and larynx to a lesser extent.

The tumor sites represented in this study are guided by the prevalence of such cancers in the centers from where samples were collected from. We did not impose any restrictions based on sites of tumor origin. The table has been updated with additional site information. (Table1).

How can the very little percentage of HPV-associated samples be explained?

HPV positivity has been shown to be very low in the patient population recruited in this study as confirmed by our collection centers. It is thus not a routinely done test as part of the initial work up or used for making treatment decisions in this region of the world.

5) The histopathologic scoring should be described in more detail (l 456)

More detailed methodology has been included in the revised manuscript (lines565-583).

6) The authors should add a paragraph placing the applicability, robustness, and validity of their model in the context of other similar 3D ex vivo models proposed in the recent literature.

A comparative table is attached below showing concordance between this platform and other 3D model platforms from two other studies.

Parameters	Mechanism of Action of test molecule			
	Stromal modulator		Myeloid reprogramming	
Platform	human breast explant histoculture	TruTumor	3D spheroid model	TruTumor
Phenotype observed	A significant increase in the infiltration of CD8+ CTLs into the tumor nest		Increase in CXCL9 secretion.	
Comments	Manuscript under review		Manuscript under preparation	

7) The authors should give an explanation why they consider a 72hr cultivation duration as sufficient to appraise therapy response.

Based on the tumor cytotoxicity response observed in SC1, we believe that in case of good responders, a 3-day culture may be sufficient to evaluate response. This period of culture might however might not be enough to address acquired resistance.

REVIEWER COMMENTS

Reviewer #1 (Remarks to the Author):

The authors have partially addressed my comments. The transparency regarding sample selection in the different analyses has improved. Furthermore, additional information regarding the optimization of the TruTumor platform and baseline samples has been included.

However, the assignment of individual genes and gene sets to functional categories still seems arbitrary and unfounded. No additional references have been included in the revised manuscript. For instance, the chemokine CCL7 is not specifically expressed by dendritic cells and why the expression of CCL7 should be associated with 'antigen presentation' is unclear to me. The same applies to additional functional categories shown in Figure 3E, e.g. 'tumor cytotoxicity'. There are little to no supporting references provided in the manuscript. Similarly, multiple of the signaling pathways listed under 'Tumor Activity' in Figure 7A play an important role in various immune cells. For instance, TGF β signaling is important in various T cell subsets.

Reviewer #3 (Remarks to the Author):

First of all I would like to express my appreciation for the efforts and time taken by the authors to substantially revise the manuscript and answer all questions.

I have only one comment left:

As the paper does not compare ex-vivo TruTumor immunotherapy response to in-patient immunotherapy responses, I would leave out 'neoadjuvant treatment given' from Table 1. Although I understand now that it was my mistake, I still find it confusing to mention clinical treatments, while these are totally not relevant for the paper.

Again, thank you for your efforts.

Reviewer #4 (Remarks to the Author):

Reviewer #4 (Remarks to the Author): with expertise in HNSCC, histocultures

Key results

The authors of the manuscript entitled Capturing the dynamic interactions between tumor and immune components of 1 the tumor micro-environment is important for determining efficacy of anti-PD1 2 treatment intend to study response to immune checkpoint inhibitors in a preclinical explant model for HNSCC. Patients could be sub-stratified on the basis of T-cell reinvigoration and tumor cytotoxicity into responders and non-responders with graduations of the features evaluated.

Validity

Your evaluation of the validity and robustness of the data interpretation and conclusions. If you feel there are flaws that prohibit the manuscript's publication, please describe them in detail.

The authors based their study on a robust and impressive case number of 70 tumor samples, The methods applied are extensive and appropriately applied. However, there are some points that need to be addressed to back the translation ability into clinical routine:

1) The samples were cultured for 72 hrs. Authors should substantiate why they chose this cultivation time. As it is accepted that resistance to anti-tumor treatments are rather long-term processes it is unclear if 72 hrs are sufficient to make a statement on treatment response.

How are early and late response defined in this regard (l 383)

2) Have the follow up data and clinical parameters of the donor patients compared with the experimental data regarding response to treatment (anti-checkpoint and other agents)?

Beyond 72 h of culture tissue dis-cohesion and pyknosis is seen. Therefore, we fixed 72 h of culture where we observed T cell activation, cytokine release and tumor killing. While

acquired resistance might not be apparent within the 3- day culture period, the platform should be able to evaluate non-response phenotype due to an immune-suppressive tumor micro-environment existing in absence of treatment.

Early and late responses were defined based on whether on treatment, just a T cell activity (e.g. Interferon gamma release) was observed versus tumor killing along with interferon gamma release respectively.

None of the patients, whose samples were included in this study have undergone treatment with anti-PD1 treatment.

3) Why were the histocultures treated with Nivolumab instead of Pembrolizumab although the CPS scores were between 3-100% in all samples?

Nivolumab was selected based on availability of cost of the drug. Given the high cost of Pembrolizumab, it is not affordable to patients in low- and middle-income countries. In a study by Tata Memorial Hospital, Mumbai, India, good response was demonstrated on treating HNSCC patients with a low dose Nivolumab and metronomic chemotherapy (Patil, V. M. et al. J Clin Oncol 41, 222–232 (2023)). We are currently running an observational trial using this treatment regimen to test the clinical validity of the TruTumor platform and develop this platform for patient selection.

COMMENT V2: The authors should add a respective paragraph explaining this procedure referring to the situation in the countries where the samples have been taken.

The findings presented here are relevant insofar as predictive markers for sub- stratifying HNSCC patients to immunotherapy are still not available. In this regards, it would be of interest to see a correlation between the follow up data of the donor patients and the experimental outcome of the explants. Did some of the donors undergo checkpoint inhibition and are there data on the respective response/ posttherapeutic follow up? As mentioned earlier, none of the patients included in this study underwent ICI treatment.

COMMENT V2: The authors should provide ongoing follow up data for their donor patients from the time point of tumor resection informing the reader about the first treatment in curative intent and first and next-line treatment in the case of metastases or recurrences. It would be of great interest if donor patients have received immunotherapy in the meantime and how they responded compared to the experimental culture data.

1) Clinical and histopathological data and features of the donor collective and the respective samples should be statistically associated with the experimental data (non-responder/responder).

Grade and stage for samples from both sub-cohorts (SC1 and SC2d) have been shown in figure 7.

2) Tumor Site Others (Table 1) should be specified as it is nearly 40% of the whole collective. Where only donor tumors originating from the oral cavity included in the trial?

3) Please explain why the proportion of oral cavity HNSCC is nearly 60%. Is there a focus on establishing a model for HNSCC of this sublocalization and what is the rationale to consider oropharynx, hypopharynx and larynx to a lesser extent.

The tumor sites represented in this study are guided by the prevalence of such cancers in the centers from where samples were collected from. We did not impose any restrictions based on sites of tumor origin. The table has been updated with additional site information. (Table1).

How can the very little percentage of HPV-associated samples be explained?

HPV positivity has been shown to be very low in the patient population recruited in this study as confirmed by our collection centers. It is thus not a routinely done test as part of the initial work up or used for making treatment decisions in this region of the world.

COMMENT V2: The authors should explain this phenomenon in the text referring to the situation in the countries where the samples have been taken.

5) The histopathologic scoring should be described in more detail (l 456)

More detailed methodology has been included in the revised manuscript (lines565583).

6) The authors should add a paragraph placing the applicability, robustness, and validity of their model in the context of other similar 3D ex vivo models proposed in the recent literature.

A comparative table is attached below showing concordance between this platform and other 3D model platforms from two other studies.

COMMENT V2: The authors should relate their 3D model to already published models instead to own unpublished data in order to make a true statement about the validity of TruTumor to allow the reader to comprehend the comparisons made.

7) The authors should give an explanation why they consider a 72hr cultivation duration as sufficient to appraise therapy response.

Based on the tumor cytotoxicity response observed in SC1, we believe that in case of good responders, a 3-day culture may be sufficient to evaluate response. This period of culture might however might not be enough to address acquired resistance.

COMMENT V2: The authors should stress this fact in the text.

RESPONSE TO REVIEWER COMMENTS:

Pointwise response provided in blue

Reviewer #1 (Remarks to the Author):

The authors have partially addressed my comments. The transparency regarding sample selection in the different analyses has improved. Furthermore, additional information regarding the optimization of the TruTumor platform and baseline samples has been included.

However, the assignment of individual genes and gene sets to functional categories still seems arbitrary and unfounded. No additional references have been included in the revised manuscript. For instance, the chemokine CCL7 is not specifically expressed by dendritic cells and why the expression of CCL7 should be associated with 'antigen presentation' is unclear to me. The same applies to additional functional categories shown in Figure 3E, e.g. 'tumor cytotoxicity'. There are little to no supporting references provided in the manuscript.

Similarly, multiple of the signaling pathways listed under 'Tumor Activity' in Figure 7A play an important role in various immune cells. For instance, TGF β signaling is important in various T cell subsets.

To simplify the gene functions we have clubbed the gene annotations into broad pathways for both figures (Fig3E and 7A). All the references based on which these broad annotations have been made are now a part of the supplementary file SF1.

Reviewer #3 (Remarks to the Author):

First of all I would like to express my appreciation for the efforts and time taken by the authors to substantially revise the manuscript and answer all questions.

I have only one comment left:

As the paper does not compare ex-vivo TruTumor immunotherapy response to in-patient immunotherapy responses, I would leave out 'neoadjuvant treatment given' from Table 1. Although I understand now that it was my mistake, I still find it confusing to mention clinical treatments, while these are totally not relevant for the paper.

The reason for including neo-adjuvant treatment was to check for effect of pre-treatment on the level of preservation post culture. We did not observe any effect of the pre-treatment on preservation. This statement has now been included in the manuscript (line: 88-89).

Again, thank you for your efforts.

Reviewer #4 (Remarks to the Author):

Reviewer #4 (Remarks to the Author): with expertise in HNSCC, histocultures

Key results

The authors of the manuscript entitled Capturing the dynamic interactions between tumor and immune components of 1 the tumor micro-environment is important for determining efficacy of anti-PD1 2 treatment intend to study response to immune checkpoint inhibitors in a preclinical explant model for HNSCC. Patients could be sub-stratified on the basis of T-cell reinvigoration and tumor cytotoxicity into responders and non-responders with graduations of the features evaluated.

Validity

Your evaluation of the validity and robustness of the data interpretation and conclusions. If you feel there are flaws that prohibit the manuscript's publication, please describe them in detail.

The authors based their study on a robust and impressive case number of 70 tumor samples, The methods applied are extensive and appropriately applied. However, there are some points that need to be addressed to back the translation ability into clinical routine:

1) The samples were cultured for 72 hrs. Authors should substantiate why they chose this cultivation time. As it is accepted that resistance to anti-tumor treatments are rather long-term processes it is unclear if 72 hrs are sufficient to make a statement on treatment response.

How are early and late response defined in this regard (l 383)

2) Have the follow up data and clinical parameters of the donor patients compared with the experimental data regarding response to treatment (anti-checkpoint and other agents)?

Beyond 72 h of culture tissue dis-cohesion and pyknosis is seen. Therefore, we fixed 72 h of culture where we observed T cell activation, cytokine release and tumor killing. While acquired resistance might not be apparent within the 3- day culture period, the platform should be able to evaluate non-response phenotype due to an immune-suppressive tumor micro-environment existing in absence of treatment.

Early and late responses were defined based on whether on treatment, just a T cell activity (e.g. Interferon gamma release) was observed versus tumor killing along with interferon gamma release respectively.

None of the patients, whose samples were included in this study have undergone treatment with anti-PD1 treatment.

3) Why were the histocultures treated with Nivolumab instead of Pembrolizumab although the CPS scores were between 3-100% in all samples?

Nivolumab was selected based on availability of cost of the drug. Given the high cost of Pembrolizumab, it is not affordable to patients in low- and middle-income countries. In a study by Tata Memorial Hospital, Mumbai, India, good response was demonstrated on treating HNSCC patients with a low dose Nivolumab and metronomic chemotherapy (Patil, V. M. et al. J Clin Oncol 41, 222–232 (2023)). We are currently running an observational trial using this treatment regimen to test the clinical validity of the TruTumor platform and develop this platform for patient selection.

COMMENT V2: The authors should add a respective paragraph explaining this procedure referring to the situation in the countries where the samples have been taken.

Editorial note: response redacted

The findings presented here are relevant insofar as predictive markers for sub-stratifying HNSCC patients to immunotherapy are still not available. In this regard, it would be of interest to see a correlation between the follow-up data of the donor patients and the experimental outcome of the explants. Did some of the donors undergo checkpoint inhibition and are there data on the respective response/posttherapeutic follow-up?

As mentioned earlier, none of the patients included in this study underwent ICI treatment.

COMMENT V2: The authors should provide ongoing follow-up data for their donor patients from the time point of tumor resection informing the reader about the first treatment in curative intent and first and next-line treatment in the case of metastases or recurrences. It would be of great interest if donor patients have received immunotherapy in the meantime and how they responded compared to the experimental culture data.

On following up we were informed that none of the 70 patients included in the study received Nivolumab treatment.

1) Clinical and histopathological data and features of the donor collective and the respective samples should be statistically associated with the experimental data (non-responder/responder). Grade and stage for samples from both sub-cohorts (SC1 and SC2d) have been shown in figure 7.

2) Tumor Site Others (Table 1) should be specified as it is nearly 40% of the whole collective. Where only donor tumors originating from the oral cavity included in the trial?

3) Please explain why the proportion of oral cavity HNSCC is nearly 60%. Is there a focus on establishing a model for HNSCC of this sublocalization and what is the rationale to consider oropharynx, hypopharynx and larynx to a lesser extent.

The tumor sites represented in this study are guided by the prevalence of such cancers in the centers from where samples were collected from. We did not impose any restrictions based on sites of tumor origin. The table has been updated with additional site information. (Table1).

How can the very little percentage of HPV-associated samples be explained?

HPV positivity has been shown to be very low in the patient population recruited in this study as confirmed by our collection centers. It is thus not a routinely done test as part of the initial work-up or used for making treatment decisions in this region of the world.

COMMENT V2: The authors should explain this phenomenon in the text referring to the situation in the countries where the samples have been taken.

This has now been addressed (Lines: 237-239)

5) The histopathologic scoring should be described in more detail (l 456)

More detailed methodology has been included in the revised manuscript (lines 565-583).

6) The authors should add a paragraph placing the applicability, robustness, and validity of their model in the context of other similar 3D ex vivo models proposed in the recent literature. A comparative table is attached below showing concordance between this platform and other 3D model platforms from two other studies.

COMMENT V2: The authors should relate their 3D model to already published models instead to own unpublished data in order to make a true statement about the validity of TruTumor to allow the reader to comprehend the comparisons made.

This has now been addressed (Lines: 349-354, Supplementary file SF3)

7) The authors should give an explanation why they consider a 72hr cultivation duration as sufficient to appraise therapy response.

Based on the tumor cytotoxicity response observed in SC1, we believe that in case of good responders, a 3-day culture may be sufficient to evaluate response. This period of culture might however might not be enough to address acquired resistance.

COMMENT V2: The authors should stress this fact in the text.

This has now been addressed (Lines: 86-88).

REVIEWERS' COMMENTS

Reviewer #1 (Remarks to the Author):

The authors have addressed my comments and I have no further objections.

Reviewer #4 (Remarks to the Author):

All queries have finally been sufficiently addressed by the authors.